# Stepwise internal potential jumps caused by multiple-domain polarization flips in metal/ferroelectric/metal/paraelectric/metal stack

Xiuyan Li [1,2 ✉] & Akira Toriumi[2]

Negative capacitance (NC) effects in ferroelectric/paraelectric (FE/PE) stacks have been recently discussed intensively in terms of the steep subthreshold swing (SS) in field-effect transistors (FETs). It is, however, still disputable to stabilize quasi-static-NC effects. In this work, stepwise internal potential jumps in a metal/FE/metal/PE/metal system observed near the coercive voltage of the FE layer are reported through carefully designed DC measurements. The relationship of the internal potential jumps with the steep SS in FETs is also experimentally confirmed by connecting a FE capacitor to a simple metal-oxide-semiconductor FET. On the basis of the experimental results, the observed internal potential jumps are analytically modelled from the viewpoint of bound charge emission associated with each domain flip in a multiple-domain FE layer in a FE/PE stack. This view is different from the original NC concept and should be employed for characterizing FE/PE gate stack FETs.

[1] National Key Laboratory of Science and Technology on Micro/Nano Fabrication, Department of Micro/Nano Electronics, School of Electronic Information and Electrical Engineering, Shanghai Jiao Tong University, Shanghai 200240, P. R. China. [2] Department of Materials Engineering, The University of Tokyo, Tokyo 113-8656, Japan. ✉email: xiuyanli@sjtu.edu.cn

Negative capacitance (NC) effects in ferroelectric (FE) films have attracted intensive attention recently in terms of both polarization kinetics modeling and low-power complementary metal-oxide-semiconductor (CMOS) applications. The original idea of NC was proposed from the fact that the Landau-Devonshire theory would possess an intrinsic NC state in a FE capacitor (CAP) with a single domain, namely an S-like polarization-voltage behavior. Although this effect is not observable in a single FE-CAP, it is expected to be stabilized by connecting a linear paraelectric (PE) CAP in series. Furthermore, since it may achieve an internal voltage amplification in gate stacks of field-effect transistors (FETs) by using the FE/PE stacks, it is very attractive that the Boltzmann tyranny (60 mV dec$^{-1}$ at room temperature) in subthreshold slope (SS) may be surmounted[1]. Such a device with no hysteresis is often called a NC-FET. In a decade, many groups have focused on studying NC effects in FE/PE stacks and FE/PE gate stack FETs[2–24]. Concerning NC effects in FE/PE stacks, three kinds of demonstrations have been provided experimentally so far: (i) total capacitance enhancement in the case of no internal metal between FE and PE layers[2–4], (ii) transient NC effects in AC mode operation[5–7], and (iii) locally stabilized NC state[8]. Several models of quasi-static NC associated with domain wall motion in a multiple-domain system have been also proposed[9–13]. Meanwhile, steep SS values have been demonstrated by incorporating FE/PE gate stacks into FETs with various FE materials[14–16], various channel materials[15,17–19] and various FET structures[14,20–24].

However, physical understanding of NC effects is still under intensive debate[25–36]. Experimentally observed NC effects are different from each other, and also from the concepts initially proposed. Alternative explanations for the capacitance enhancement and transient NC effects in FE/PE stacks have been also proposed[25–31]. For example, a feasibility of capacitance enhancement is explainable from a strong coupling between FE and PE layers[25–27], while the transient NC is understandable from the viewpoints of overshoot in voltage supply or slower speed of charge compensation relative to polarization switching in FE-CAP[28–31]. In fact, it has been argued that NC region of FE material is intrinsically unstable or even impossible[32,33]. In addition, the SS improvements observed in FETs mostly suffer from critical problems that a large hysteresis is detected, a high voltage is needed and an operation frequency is limited in actual experiments[34–36].

To sum up, the experimental evidences provided so far are insufficient to conclude the concept of quasi-static NC, and reliable modeling of SS improvement in a FET with FE/PE gate stack is still missing. These should be verified urgently, because they are critical for further advancing the material science as well as electron device performance of FE/PE stacks to elucidate whether the quasi-static NC can be really stabilized or not, and whether the steep SS characteristics so far demonstrated are really promising for low-power CMOS applications or not. A direct way to examine the actual FE effect in FE/PE stack is to investigate the voltage at the internal node, $V_{int}$, between FE and PE layers in DC mode, which makes possible of the direct correlation of $V_{int}$ with SS in FET. In fact, a couple of works on internal potential measurement have been reported, but they have only qualitatively discussed about this issue, resulting that a consistent model could not been provided[16,37]. We have suspected it might be due to experimental difficulties of measuring the internal potential in FE/PE stack. Therefore, in this work, accurate DC measurements are particularly paid careful attention. The stepwise $V_{int}$ jumps at the coercive voltage, $\pm V_C$, of FE layer in FE/PE stack are demonstrated, and a relationship between $V_{int}$ jumps and the steep SS in FET with FE/PE gate stack is presented. They are quantitatively understood from the viewpoints of successive polarized domain flipping and depolarization feedbacks from the PE-CAP. The results provide a clear physical insight to understanding the small SS values in FETs with FE/PE gate stacks reported so far.

## Results

**DC measurement of internal potential in FE/PE stack.** To accurately estimate $V_{int}$ in FE/PE stack in DC mode, a PE/PE stack was firstly inspected quantitatively. In an ideal case, total charges at the internal metal, which is electrically floating, have to be conserved as long as no leakage current through both CAPs is assured (Fig. 1a). However, since the DC output impedance at the floating node is infinite in the ideal case, it is substantially impossible to estimate $V_{int}$ experimentally. In actual cases, a finite resistance of capacitors enables to measure $V_{int}$ in DC-mode, while a time constant should be considered for a quantitative analysis. More importantly, a small but a finite amount of charge-flow from the internal metal to a measurement system should also be paid attention to, because the measured voltage in itself is significantly affected by the input impedance in the measurement system. Thus, the actual equivalent circuit concerned in this work is shown in Fig. 1b. Based on the formulation shown in Supplementary Note 1[38] and the measured resistance of PE CAPs (~$10^{12}\,\Omega$), a voltage measurement system with the input impedance higher than ~$10^{14}\,\Omega$ was needed for accurate DC-mode measurements. We used a high input impedance voltmeter equipped with a high precision current preamplifier, which was able to adjust a voltage to suppress a current flow down to zero (sub-fA level). By doing so, the effective input impedance was enhanced up to $10^{16}\,\Omega$. Figure 1c compares $V_{int}$ in PE/PE stacks measured in the present system and in conventional voltmeter with an input impedance of ~$10^{10}\,\Omega$. When the present system was employed, the accurate measurement of $V_{int}$ was successfully assured for capacitors with down to 1 pF. Namely, it is critically important for characterizing the $V_{int}$ quantitatively to take the input impedance of the measurement system into account.

In actual measurements of $V_{int}$ in FE/PE stacks, commercially available PZT films with Pt electrodes were used as the FE-CAPs. The typical charge–voltage ($Q$-$V_F$), capacitance–voltage ($C_F$-$V_F$) and leakage ($I$-$V_F$) characteristics are shown in Supplementary Fig. 1. The capacitance at the center of $C_F$-$V_F$ characteristics was ~0.25 nF and the leakage current density at 1 V was below ~$10^{-8}$ A cm$^{-2}$, which is comparable to the level reported for the state of the art PZT[39,40] and corresponds to $5 \times 10^{10}\,\Omega$. The equivalent circuit of the $V_{int}$ measurement in FE/PE system shown in Fig. 2a was assumed, in which impedances in both PE-CAP and measurement system were much higher than that in PZT. In prior to measurements, the FE-CPA was polarized by a negative voltage, and the internal terminal of FE/PE stack was grounded to remove the unknown charges stored. More details in the measurements are described in the method section. Figure 2b and c show $V_{int}$ and $\delta V_{int}/\delta V$ in sweepings of the total voltage, $V$, in the case with PE capacitance ($C_P$) of 0.5 nF. A big $V_{int}$ jump is seen in both forward and backward $V$ sweepings, and each of them corresponds to the differential gain of $V_{int}$ ($\delta V_{int}/\delta V > 1$). It is noted that the big $V_{int}$ jump is followed by successively oscillating small ones as shown in enlarged part of Fig. 2b. $V_F$ calculated by subtracting measured $V_{int}$ from $V$ is shown as a function of $V$ in Fig. 2d. Note that the $V_F$ drops along with the $V_{int}$ jumps occur very near $\pm V_C$ in FE-CAP. It directly indicates that the $V_{int}$ gain is associated with the polarization flip in a FE layer. This fact is critically important from the viewpoint that it is beyond the qualitative observation in the similar measurements reported recently[16,37]. It is worthy of mentioning that the measured voltage was time dependent and it gradually changed

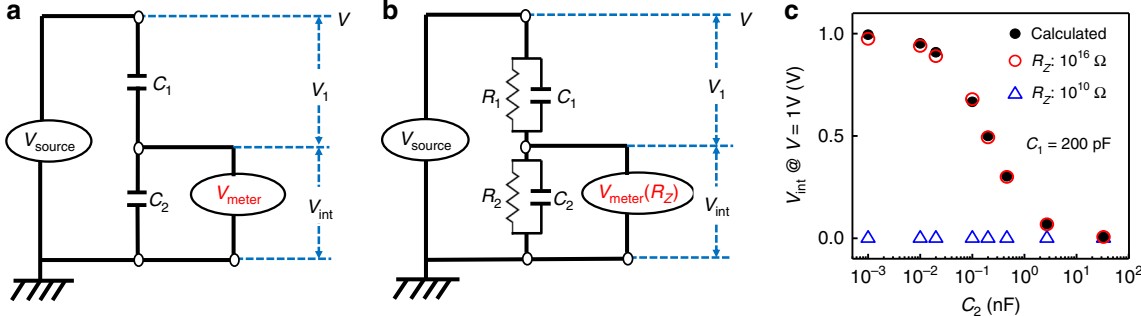

**Fig. 1 Understanding and investigation of $V_{int}$ in PE/PE stacks. a** Ideal and **b** actual equivalent circuits in the internal potential measurement in PE/PE stacks. $C_1$/$C_2$ and $R_1$/$R_2$ are the capacitance and insulating resistance of each FE-CAP. $V_1$ is the voltage applied on the top PE-CAP. $R_Z$ is the input impedance of $V_{int}$ measurement system. **c** $V_{int}$, with $V = 1$ V and $C_1 = 200$ pF, as a function of $C_2$ from 1 pF to 33 nF for two cases using high input impedance system and conventional DC voltmeter. Only high impedance system enables to estimate the internal potential in DC mode measurement.

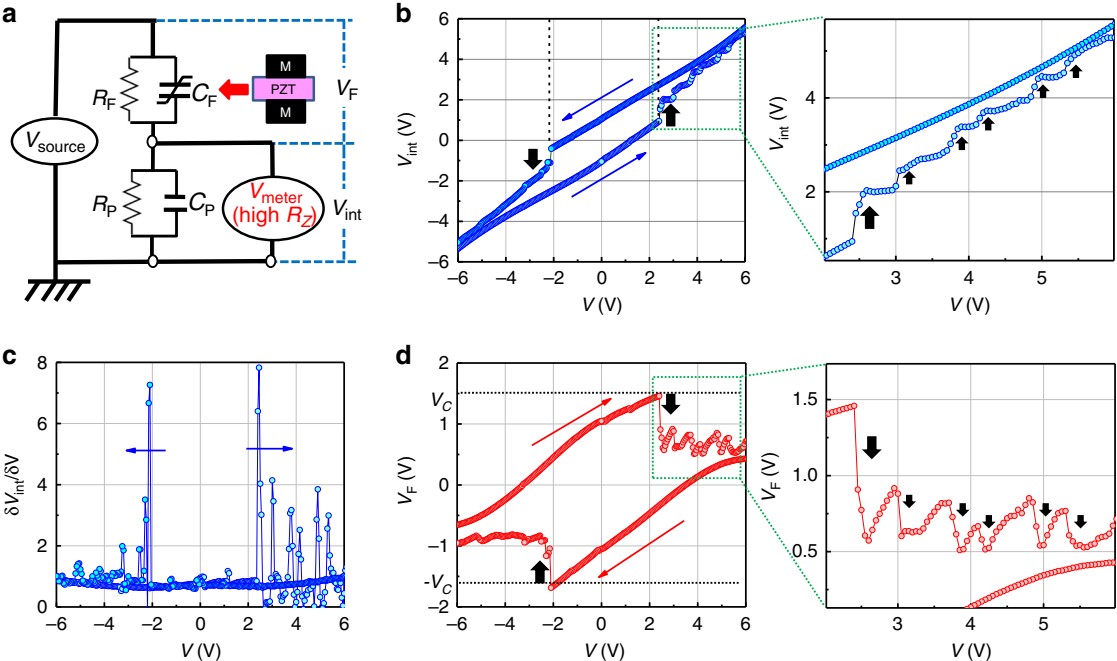

**Fig. 2 $V_{int}$ measurement in a FE/PE stack. a** Equivalent circuit of $V_{int}$ measurement in FE/PE system. $R_F$ and $R_P$ are the insulating resistance of FE- and PE-CAP respectively. Note that high impedance system is needed to get the accurate $V_{int}$. **b** $V_{int}$-$V$, **c** $\delta V_{int}/\delta V$-$V$ and **d** $V_F$-$V$ characteristics during $V$ sweeping of FE/PE system. $V_{int}$ jump occur along with $V_F$ drop at $V_F = \sim\pm V_C$, corresponding to $\delta V_{int}/\delta V > 1$, followed by the oscillating small ones.

due to the finite resistance in FE and PE layers as mentioned above. Therefore, the absolute value of $V_{int}$ is not as expected in $C_P$-$C_F$ circuit. Both $V$ and $V_{int}$ are, however, accurately measurable and $V_F$ obtained by $V$-$V_{int}$ should also be accurate quantitatively, as long as the input impedance of voltmeter is effectively higher than the output impedance in the PE-CAP and FE-CAP and the measurement time is substantially smaller than the time constant of the FE/PE system. This is further confirmed by the results in Supplementary Fig. 2, that the experimental observations are well reproduced by repeating the measurement for three times and the $V_F$ positions corresponding to $V_{int}$ jumps remain to be near $\pm V_C$ even by changing $C_P$ from 15 nF down to 0.1 nF. In addition, a possible origin of the $V_{int}$ variation in FE/PE system might be the time constant variation originated from the bias dependence of resistance and/or capacitance of FE-CAP. But, such a concern is irrelevant to the present experimental observations, because the results are not affected at all even in the FE/PE stacks with different time constants (with different $C_P$ values).

**Correlating internal potential gain with steep SS in FET.** A relationship of the internal potential gain with the SS improvement in FET operation is next investigated. A FE-CAP was connected through a cable to a poly-Si gate/SiO₂ metal-oxide-semiconductor FET (MOSFET) with a channel width, length, and oxide thickness of 200 μm, 500 μm, and 5 nm, respectively. Accumulated MOS capacitance was ~0.7 nF and gate leakage was ~$10^{-9}$ A cm$^{-2}$ at 1 V. The equivalent circuit is shown in Fig. 3a. The source-drain current ($I_{DS}$) and $V_{int}$ of the FE-CAP/MOSFET stack were measured separately in gate bias ($V_{GS}$) sweeping under drain voltage ($V_{DS}$) of 0.1 V. It is mandatory to eliminate unknown charge effects at the internal node in FE/MOSFET prior to each measurement, because floating charges are very likely to affect the charge dynamics in the domain reversal. This is critical for considering the hysteresis in $I_{DS}$-$V_{GS}$ characteristics of FE/MOSFET system. Figure 3b shows $I_{DS}$-$V_{GS}$ curves *w/* and *w/o* connecting the FE-CAP. The SS in the case *w/* FE-CAP clearly looks sharper than that in *w/o* case in a given $V_{GS}$ region. Figure 3c plots $V_{GS}$ dependence of SS. In the case *w/* FE CAP, SS is

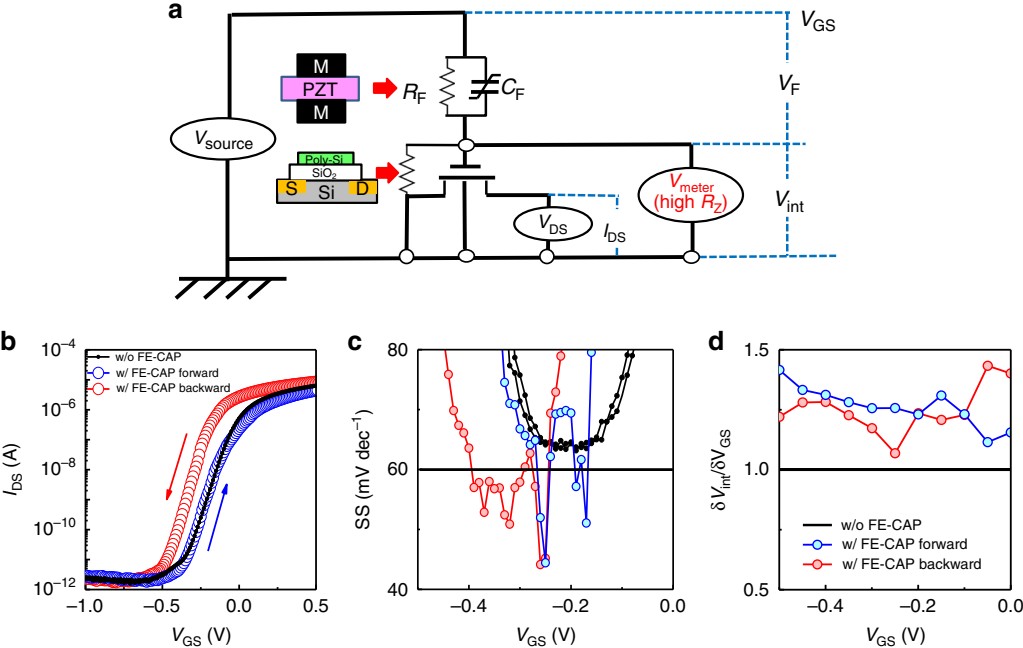

**Fig. 3 Correlation of $V_{int}$ gain with steep SS in FET. a** Equivalent circuit for measuring $I_{DS}$-$V_{GS}$ and $V_{int}$-$V_{GS}$ characteristics in FE/MOSFET system. Note that $V_{int}$-$V_{GS}$ and $I_{DS}$-$V_{GS}$ were measured separately for the same system and under the same $V_{DS}$ (0.1 V). **b** $I_{DS}$-$V_{GS}$, **c** SS-$V_{GS}$ and **d** $\delta V_{int}/\delta V_{GS}$-$V_{GS}$ characteristics of MOSFET *w/* and *w/o* external FE-CAP during sweepings of $V_{GS}$. A correlation between sub-60 mV dec$^{-1}$ SS and internal potential gain is demonstrated.

improved down to sub-60 mV dec$^{-1}$ both in forward and backward sweepings. $\delta V_{int}/\delta V_{GS}$ as a function of $V_{GS}$ is shown in Fig. 3d. Note that the improvement factor of SS (a ratio of SS *w/o* to *w/* FE-CAP) is roughly the same as $\delta V_{int}/\delta V_{GS}$ in Fig. 2d. These results indicate that the steep SS value below the Boltzmann limit is definitely associated with $V_{int}$ gain in FE/MOSFET system.

**Electrostatic understanding of internal potential gains**. Next, a possible kinetic origin of stepwise $V_{int}$ jumps is discussed. The charge dynamics associated with the domain flipping in FE/PE stack is a main focus. We suppose the capacitance coupling circuit in FE/PE stack for characterizing internal charge kinetics in the following.

It is here noted that two kinds of charges are involved in FE-CAP: bound charges ($Q_{bound}$) and free charges ($Q_{free}$), as shown in Fig. 4a, where the positive polarization, $P$, is defined as the arrow from left to right. If FE-CAP is pre-polarized by a negative voltage as that in the experiments, $Q_{bound} = -P$ initially. According to electrostatics, the relevant system should satisfy the following equations before the polarization switching:

$$V_F + V_{int} = V \qquad (1)$$

$$V_{int} = \frac{Q}{C_P} \qquad (2)$$

$$Q = Q_{free} + Q_{bound} = C_F V_F - P \qquad (3)$$

in which $Q$ denotes total charges accumulated on each CAP. Assuming $C_F$ is constant, $V_{int}$ and $V_F$ are expressed by

$$V_{int} = \frac{C_F}{C_F + C_P} V - \frac{P}{C_F + C_P} \qquad (4)$$

$$V_F = \frac{C_P}{C_F + C_P} V + \frac{P}{C_F + C_P} \qquad (5)$$

Since the bound charges are fixed in the case without polarization flipping, the FE/PE stack is similar to a PE/PE one. Thus,

$$\frac{\delta V_{int}}{\delta V} = \frac{C_F}{C_P + C_F} < 1 \qquad (6)$$

On the other hand, when the polarization flipping occurs at $V_F = \sim V_C$ (taking the forward sweeping as an example), the polarization, $P$, should change the direction in Eqs. (3), (4) and (5). Namely, a bunch of charges are transferred from FE-CAP to PE-one. If we assume FE layer is with a single domain,

$$\frac{\delta V_{int}}{\delta V} = \frac{C_F}{C_P + C_F} + \frac{\frac{2P}{\delta V}}{C_P + C_F} \qquad (7)$$

This indicates that $\delta V_{int} > \delta V$ can become to be >1 in case with a relatively small $C_P$. Namely, $V_{int}$ jump along with $V_F$ drop should occur in this case.

In actual FE films, multiple domains are involved and not all domains flip simultaneously. Here, it is reasonably assumed that each domain flips independently and that it has each $V_C$ with a tight distribution in FE film. When a certain domain flips with a relatively small $C_P$, $V_F$ should be reduced due to a finite amount of charge transfer from FE-CAP to PE-one. It means that remaining domains cannot flip before further increase in $V_F$ to $V_C$. When $V_F$ is increased to $V_C$ again, another domain flips, followed by the $V_F$ drop again. Thus, the initial $V_{int}$ jump and $V_F$ drop followed by oscillating $V_{int}$-$V$ and $V_F$-$V$ characteristics near $V_C$ are clearly explained. The stepwise $V_{int}$ jumps are analytically described as

$$\frac{\delta V_{int}}{\delta V} = \frac{C_F}{C_F + C_P} + \frac{\frac{2P_i}{\delta V}}{C_F + C_P} \qquad (8)$$

in which $P_i$ ($i = 1, 2, 3...$) is the polarization of each domain in a FE film. Since the $V_F$ drop means the depolarization field formation in FE-CAP, each $V_{int}$ jump can be regarded as a depolarization feedback from PE-CAP associated with each

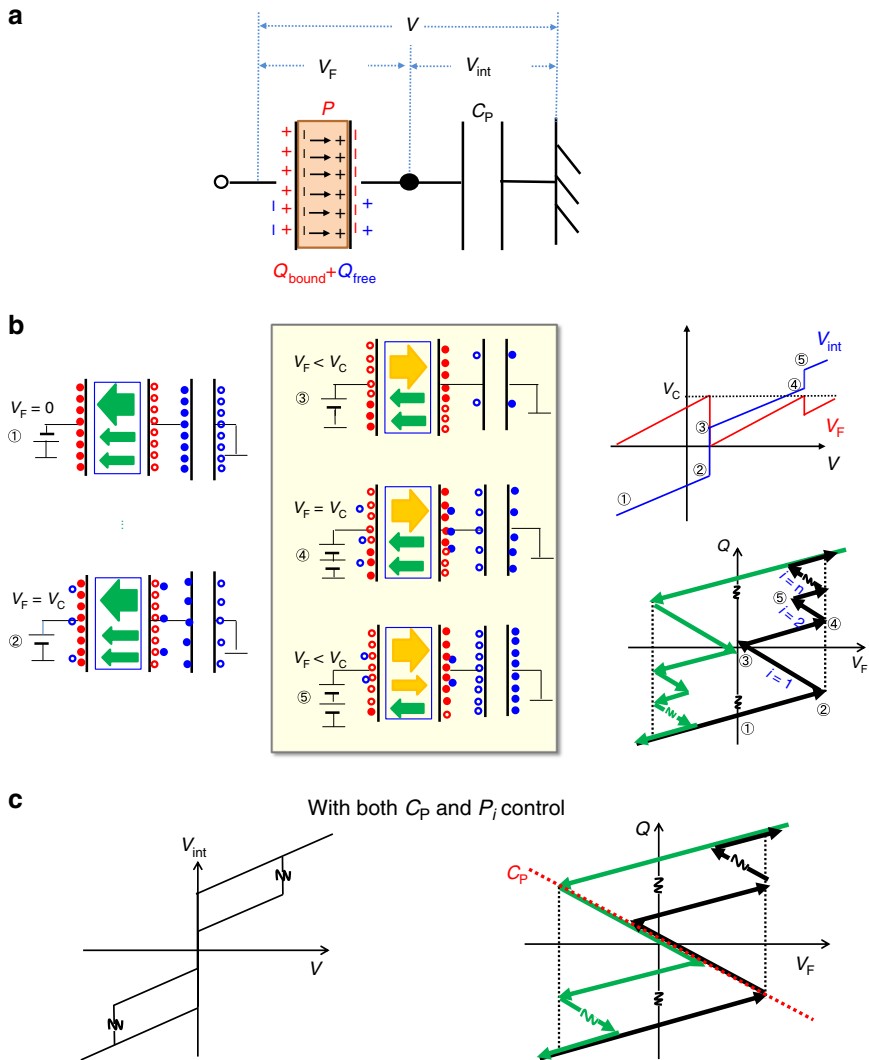

**Fig. 4 Electrostatic understanding of stepwise $V_{int}$ jumps in FE/PE stacks. a** Schematics of FE/PE system. Two kinds of charges are involved in FE-CAP: bound charges ($Q_{bound}$) and free charges ($Q_{free}$). **b** Understanding of charge dynamics (circles and dots represent holes and electrons respectively), voltage change and $Q$-$V_F$ characteristics in FE/PE system with successive multiple-domain flips. **c** A possible case that the biggest $V_{int}$ jump in the forward $V$ sweeping may overlap partly with that in the backward one, then the subthreshold hysteresis in FE/PE gate stack FET may be much reduced in the overlapping $V_{int}$ range if the bias dependence of semiconductor is ignored.

domain flip. Namely, successive multiple-domain flips accompanying the depolarization feedbacks from PE-CAP result in zigzag $Q$-$V_F$ characteristics. These views are schematically shown in Fig. 4b in detail. Note that the resulted zigzag $Q$-$V_F$ characteristics is totally different from the conventional $Q$-$V_F$ curve in a single FE-CAP or from the S-curve expected from the original NC theory[1]. It is also significantly different from the characteristics expected from the recent models with the multiple-domain system, in which a continuous change of polarization with the help of domain wall motion is assumed[9–13].

Hysteresis of the $V_{int}$ gain is critical for achieving steep SS FETs in advanced CMOS. According to our modeling, nearly hysteresis-free $V_{int}$ gain is made possible in principle by inserting an appropriate $C_P$ in the case of the single domain FE in FE/PE stack. The requirement for this condition is the same as that for stabilizing NC effect originally proposed. This is also partly consistent with a recent report in which nearly single domain PZT film together with strict capacitance matching was employed and a nearly hysteresis-free steep SS was demonstrated[23,24]. In the case of FE layer containing multiple domains, the hysteresis-free $V_{int}$ gain is difficult to achieve perfectly. Here, let us focus on

the biggest $V_{int}$ gain. Since the $V$ value and amplitude of this $V_{int}$ jump are controlled by $C_P$ and $P_1$, respectively, it becomes possible that the biggest $V_{int}$ jump in the forward $V$ sweeping may overlap with that in the backward one by adjusting $P_1$ under an optimum $C_P$, (Fig. 4c). Resultantly, the subthreshold hysteresis in FETs should be substantially reduced in the overlapped range. As a matter of fact, the bias dependence of semiconductor capacitance makes hysteresis-free FET more difficult. Thus, hysteresis is seen in most of the steep-SS FETs reported up to now[16,22,23]. More detailed discussion of the hysteresis is shown in Supplementary Fig. 3 and Supplementary Note 2.

In addition, it should be mentioned that the domain–domain interaction is needed to take into consideration and the internal field inside the film may be reduced in the first domain switching in the real multiple-domain system. These effects should affect the voltages for successive domain flipping. In fact, our results in Fig. 1d show that the successive domain flips, following the biggest one, occur at $V_F$ values slightly smaller than $V_C$.

**Difference between present and original views on NC effect.** The intrinsic difference between our view and the original

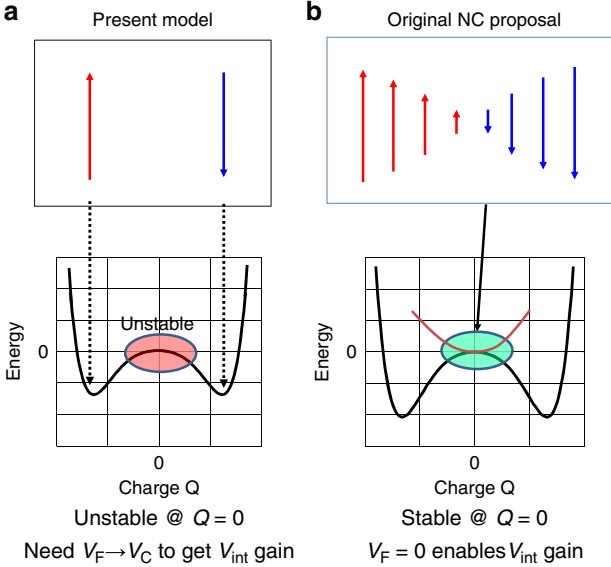

**Fig. 5 Comparison of present view with initial NC proposal. a** In present view, $V_{int}$ gain is associated with polarization flip, and the intermediate state between two polarization states is unstable. Thus, $V_F > V_C$ is required to get $V_{int}$ gain. **b** In the original NC proposal, the intermediate state can be stabilized by changing the polarization value. Then even $V_F = 0$ enables a $V_{int}$ gain directly.

proposal on NC effects in FE-based FETs is next discussed. To be fair, only the case of single domain is considered. The $V_{int}$ gain is actually obtained in both cases. In the present view, it is associated with the domain flip, on the assumption that an intermediate state between two polarized states in the domain is not stable quasi-statically (Fig. 5a). This requires that the gate voltage should be high enough to enable $V_F > V_C$. While in the original NC proposal, the intermediate state with a small polarization value can be stabilized with a small $C_P$, and the $V_{int}$ gain is obtainable even at $V_F = 0$ (Fig. 5b). It remains still unclarified whether it is possible to access to the intermediate polarized state quasi-statically or not. It is, however, strongly inferred that the continuous change of polarization in a domain is very improbable from the present experimental results. This is also consistent with the fact that no observation of the SS improvement under a small gate voltage swing has been reported so far.

Finally, it is worthy of mentioning that the polarization switching kinetics depends on a specific model such as Kolmogorov-Avrami-Ishibashi or nucleation-limited switching models[31,41], while specific switching kinetics does not necessarily lead to the total frustration of the $V_{int}$ enhancement effect[12]. Namely, when the depolarization field is formed due to the bound charge movement, the $V_{int}$ gain can be obtained in any switching kinetics cases. Meanwhile, the gain value, shape and time dependence of $V_{int}$, might be affected. But such consideration is beyond the scope of this paper in which the DC-mode polarization switching rather than high-speed switching is focused. When the polarization switching speed is concerned in device applications, the specific domain switching kinetics should be taken into account for the NC effect analysis. In addition, in the case of multiple-domain system, there will be a difference between the cases w/ and w/o internal electrode. In the latter case, the charge flow at FE/PE interface, the local effect of domain switching and the coupling effects between FE and PE layer should be taken into consideration[11,25–27]. It is very interesting to further study these issues.

## Discussion

This work demonstrates stepwise internal potential jumps associated with successive domain flips in multiple-domain FE/PE stack, and a direct relationship of this effect with steep SS in a SiO$_2$ MOSFET connected with a FE capacitor. Each domain flip provides an internal potential gain in FE/PE stack with a suitable PE capacitance as originally proposed NC effect, while a stable intermediate state between two polarization states is very unlikely. Thus, the power supply voltage surmounting the coercive voltage on FE layer is required, which is not advantageous for low-power device applications. The nearly hysteresis-free steep SS FET may be possible with a single-domain FE layer by tuning PE capacitance, while that with a multiple-domain FE layer is very challenging. It is likely that most of small SS values so far reported in literatures are explainable by the present model.

## Methods

**Measurement circuits of FE/PE and FE/MOSFET systems**. PZT-CAPs with Pt electrodes available from Radiant Technologies Inc. and conventional PE-CAPs were used. FE and PE CAPs were connected in series by a coaxial cable. A DC-voltage measurement system (Keithley 6430) equipped with a remote sub-fA pre-amplifier was connected to the internal metal node between FE and PE CAPs to measure the internal potential. This enabled us to effectively increase the input impedance of the measurement system up to 10$^{16}$ Ω. For FE/MOSFET system, poly-Si gate/SiO$_2$/Si MOSFET was connected to PZT-CAP in series. The FE/MOSFET system was connected to a semiconductor parameter analyzer (Keysight B1500) for $I_{DS}$-$V_{GS}$ characterization, while the high impedance system was connected to the internal metal between FE and MOSFET for internal potential measurement. Both experiments were carried out separately under the same $V_{DS}$.

**Electrical measurement**. Before measurement, each terminal in FE/PE and FE/MOSFET circuits was grounded to remove unknown charges left inside. Then the internal potential, $V_{int}$, was measured by sweeping the total voltage, $V$, with a step of 50 mV. In each step, a waiting-time of 0.1 s was set for stabilizing the voltage and a total time of 0.5 s was taken for each point measurement. For the FE/MOSFET system, in addition to $V_{int}$ measurement, FET characteristics were measured with the source grounded under $V_{DS} = 0.1$ V. The sweeping step of $V_{GS}$ was 20 mV and the waiting-time for each step measurement was 0.1 s.

## Data availability

The experimental data in the present study are available from the corresponding author upon reasonable request.

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

## Acknowledgements
This work was supported by JST-CREST (JPMJCR14F2) in Japan and partly by National Natural Science Foundation of China (61904103, 91964110) and Shanghai Science and Technology Innovation Action Plan (19ZR1475300, 19JC1416700) in China. We would thank T. Nishimura and T. Yajima at The University of Tokyo for discussion and suggestions.

## Author contributions
A.T. conceived the idea of internal potential measurement in FE/PE and FE/MOSFET systems and supervised the experiments. X.L. performed circuit setup, the electrical measurement and the data analysis. X.L. and A.T. proposed the physical model and summarized the manuscript.

## Competing interests
The authors declare no competing interests.
