## [Peer Review File · Nature Communications]

Review on the manuscript “Stepwise internal potential jumps caused by multiple-domain polarization flips in metal/ferroelectric/metal/paraelectric/metal stack” by Xiuyan Li, and Akira Toriumi.

In this manuscript, the authors designed the metal/ferroelectric/metal/paraelectric/metal (MFMIM) circuit, and the metal/ferroelectric/metal/oxide/semiconductor FET circuit to perform the experiment on the transient voltage boosting effect from the PZT ferroelectric capacitor. The authors confirmed the peculiar zigzag shape of the transient negative capacitance during the reversal of the ferroelectric polarization near V_c . It is very interesting that the transient negative capacitance emerges in this stepwise manner, and the authors explanation that the successive voltage drops can be induced from the multiple domain flips due to the multi-domain structure of the ferroelectrics seems reasonable. Moreover, their suggestion of using high impedance to measure the internal voltage seems critical to perform this type of experiment without involving any inaccuracy. However, some of their critical suggestions based on their electrostatic model seem inaccurate and also inconsistent with previously reported experimental results. Therefore, these inconsistencies should be corrected or resolved by taking consideration of the following comments:

1. As noted by authors, the charge dynamics of the domain flips in FE should be taken into account in order to understand the physical origin of the transient negative capacitance. In fact, there have been already two major perspectives to this phenomenon; one is to interpret this behavior as the manifestation of the intrinsic negative capacitance of the Landau barrier as originally suggested by Khan’s paper [*Nat. Mater.* 2015, 14, 182]. The other is to treat them as an artefact of complex nonlinear system involving the classical switching kinetics of the ferroelectrics [*J. Appl. Phys.* 2018, 123, 105102]. (Also, in this regard, we highly recommend the authors to refer to another critical study from Kim et al. [*Nano Lett.*, 2017, 17, 7796] which is not cited in this manuscript) We believe that the authors’ view is more related to the latter perspective based on their statement that the polarization flip is likely to be involved in the SS improvement, and that the continuous change of polarization is very improbable. Therefore, in the discussion, the section IV of this manuscript, we recommend the authors to clarify that there have been two different perspectives to this behavior, and the results of this experiment also strongly support the latter interpretation.
2. The electrostatic model proposed by the authors in the section III seems to have an inaccurate expression or inadequately reflect the physics of the domain flips. First of all, we believe that the equation (4) and (5) seems wrong, or at least misleading. Starting from the total charge density $Q = Q_{free} + Q_{bound} = C_F V_F + P$ (i.e. eqn. (3)) and the Kirchhoff’s voltage law, each voltage, V_{int} and V_F should be obtained as follows,

$$V_{int} = \frac{C_F}{C_P + C_F} V + \frac{P}{C_P + C_F}$$
$$V_F = \frac{C_P}{C_P + C_F} V - \frac{P}{C_P + C_F}$$

Therefore, the sign of the second term in the equations should be reversed. This inaccurate expression leads to unnecessary confusion associated with equation (7) because the sign of the second term in this equation become reversed with respect to equation (4).

More importantly, the equation (7) seems to be wrong in the first place. The internal voltage boosting factor, $\frac{\delta V_{int}}{\delta V}$ can be directly obtained by differentiating V_{int} the equation (4) (with the correct sign) with respect to V as follows,

$$\frac{\delta V_{int}}{\delta V} = \frac{C_F}{C_P + C_F} + \frac{1}{C_P + C_F} \frac{\delta P}{\delta V}$$

However, the equation (7) is not correct since the last term does not involve the derivative of the bound charge P with respect to the applied voltage V , but just the switching amount of the polarization ($2P$). We believe this wrong expression of the equation (7) results in a serious inconsistency with the actual physics. Unlike the explanation from this model, the voltage boosting is attributed to the fast switching rate ($\frac{\delta P}{\delta V}$) near (or above) V_c . As explained by Saha's model [*J. Appl. Phys.* 2018, 123, 105102], the overshooting voltage above V_c without the domain switching leads to the abrupt voltage drop ($\frac{\delta V_F}{\delta V} < 0$) once the retarded switching is initiated. However, the rate of the switching $\frac{\delta P}{\delta V}$ is obviously highly nonlinear function of ferroelectric voltage, which is abruptly increased near V_c . Therefore, since the amount of the voltage drop is not determined by the magnitude of P but the switching rate $\frac{\delta P}{\delta V}$, it is highly improbable that the transient NC is maintained near $V \sim 0$ (far below the coercive voltage) where $\frac{\delta P}{\delta V}$ is very small. This is why the transient negative capacitance in the ferroelectric capacitor emerges near the coercive voltage. Although it seems reasonable that the stepwise voltage drop occurs due to the successive domain flips, it should be noted that even the zig-zag voltage drop after the biggest overshoot voltage is mainly concentrated near the coercive voltage (~ 0.6 V and ~ -1 V) of the ferroelectric as shown in Fig. 1(e).

3. We believe that the hysteresis engineering scheme proposed by the authors in the section 3 of the supplementary manuscript is also not free from the inconsistency of their models we addressed above. As a matter of fact, the modulation the capacitance of the external serial capacitor has been experimentally tested by Khan et al. [*Appl. Phys. Lett.* 2017, 111, 253501] Their results (Fig.3 of the Khan's paper) showed that the decrease of the external capacitor does not necessarily lead to the hysteresis free (quasi-static) operation of the ferroelectric capacitor. As shown in Fig.3, a less steep load line of a smaller capacitance leads to a decrease of the amount of the domain reversal, not a decrease of the magnitude of the hysteresis. This is because the transient voltage drop only emerges where $\frac{\delta P}{\delta V}$ is large enough, which is near the coercive voltage.

Reviewer #2 (Remarks to the Author):

This work is dedicated to studying and modeling internal potential jumps like a stepwise near the coercive voltage. The authors A relationship of this effect with the steep SS in FET is also demonstrated experimentally by connecting a FE capacitor to a simple metal-oxide-semiconductor FET.

1) This zig-zag effect in polarization is not new at all and overall, it is a too focused technical work to be publishable in Nature Communications. To be very clear on the novelty, the experimental proof of multiple jumps due to multiple ferroelectric domains has been previously reported by A. Saeidi et al, 'Negative capacitance as digital and analog performance booster for complementary mos transistors,' Publication date 2018/4/25, Journal arXiv preprint arXiv:1804.09622 (now published in Scientific Reports), and also showed in various conference talks. The authors seem not to be aware or they just skip such obvious literature available for polycrystalline PZT used in similar experiments, which is not acceptable.

2) Even if the modeling part seems to have some originality, this is not enough for publication in Nature Communications, therefore it should be submitted to a more technical journal. However, even in the modeling part the authors speculate that every domain flip could enables an internal potential gain in FE/PE stack. The model associated with the domain flip, on the assumption that an intermediate state between two polarized states of each domain is not stable, is put in contrast with the NC original effect (Fig. 4). Apart the multi-domain interactions that are correctly addressed by the authors, even the full model is not new, as it is known that many authors wrongly reported steep-slopes in transistors due to polarization switching and not due to true NC effect. Therefore, the report made and the results are not really surprising for an expert and are not of practical use in low power transistors/circuits, as the authors themselves are recognizing in the conclusion section.

3) The experiment itself provides useful data but it is not original and, as setup. was part of many publications in the past, many not properly cited by the authors. The authors are not even carefully considering or discussing the parasitics related to such experiment.

In conclusion, I cannot see how this work can be published in Nature Communications and should be redirected to another IEEE journal.

Reviewer #3 (Remarks to the Author):

This paper advances the understanding of the so-called "negative capacitance" phenomena that have attracted a great deal of interest in the last few years. The authors treat the multi-domain ferroelectric thin film under discussion as a combination of small Landau domains, with each domain following the Landau theory of ferroelectricity. This approach enables the authors to successfully explain those sub-60 SS data often reported for I_d - V_g measurements resulting from switching transients. Instead of adopting the popular "quasi-static negative capacitance" argument, the authors show convincing experimental data and clearly state that the apparent "negative capacitance" effect only occurs during ferroelectric switching for the multi-domain FE, which requires $V_g > V_c$. This is very important contribution that deserves quick publication to educate the readers who may very well have been exposed to papers in the literature that erroneously claim the observation of "quasi-static negative capacitance".

However, there are some minor flaws in the description of the experimental setups and the details of the measurements, which need clarification before this paper is published.

1. The equivalent circuits drawn in Fig. 1(a) and Fig. 2(a) give the impression that the M/PZT/M sample being measured is represented by a capacitance without any leakage current. In reality, any realistic M/PZT/M sample has a finite leakage current, which should be represented by a parallel resistance (which is voltage dependent) across the M/PZT/M sample. Actually, in the "Supplementary Material" Section, Fig.S1(b) depicts the more appropriate equivalent circuit, which should be used in the main manuscript..
2. For the measurement setup, the authors emphasize the importance of the 'high-Z' voltage meter, and set the input impedance of the voltage meter to be 10¹⁶ ohms. The authors should explain in detail how this is accomplished.
3. In Fig. 1 (b), when the applied voltage is +/-6V, the internal voltage is around +/- 5V. Since according to the manuscript, CFE is 0.25nF and CP is 0.5nF, the measurement results indicate that the voltage divider is not based on the capacitances but rather the impedances of the two capacitors, which is exactly why the previous comment #1 was made by the reviewer. The authors need to clarify this. In connection with this, the authors should explain in detail how the DC measurement of Fig. 1 (b) is performed.
4. In Fig. 3, the authors assume that all the small domains have the same coercive field, while in reality, the ferroelectric domains may have different orientations in a given sample. Since the applied voltage induces an electric field (E_{app}) perpendicular to the electrodes, some domains may require $E_{app} > E_c$ to switch due to the domain mis-alignment. Therefore, it is more realistic to assume that the effective coercive field has a distribution instead of one identical value. The authors are urged to take this into consideration in their revision.
5. Concerning Fig. 2, prior to the I_d - V_g measurement, the authors should state whether they discharged the ferroelectric capacitor and the internal node. The reason for this question is that when an external capacitor is connected to the gate, the threshold voltage may shift to the right (for NMOS). If the internal node has finite charge, then the threshold voltage may change differently. This charge may also affect the SS results. The authors state that in the FE-PE measurement, the internal node is discharged, but for the FE-MOS measurement, this is not mentioned. Please clarify this.

Reviewer #1

Review on the manuscript “Stepwise internal potential jumps caused by multiple-domain polarization flips in metal/ferroelectric/metal/paraelectric/metal stack” by Xiuyan Li, and Akira Toriumi.

In this manuscript, the authors designed the metal/ferroelectric/metal/paraelectric/metal(MFMIM) circuit, and the metal/ferroelectric/metal/oxide/semiconductor FET circuit to perform the experiment on the transient voltage boosting effect from the PZT ferroelectric capacitor. The authors confirmed the peculiar zigzag shape of the transient negative capacitance during the reversal of the ferroelectric polarization near V_C . It is very interesting that the transient negative capacitance emerges in this stepwise manner, and the authors explanation that the successive voltage drops can be induced from the multiple domain flips due to the multi-domain structure of the ferroelectrics seems reasonable. Moreover, their suggestion of using high impedance to measure the internal voltage seems critical to perform this type of experiment without involving any inaccuracy. However, some of their critical suggestions based on their electrostatic model seem inaccurate and also inconsistent with previously reported experimental results. Therefore, these inconsistencies should be corrected or resolved by taking consideration of the following comments:

1. As noted by authors, the charge dynamics of the domain flips in FE should be taken into account in order to understand the physical origin of the transient negative capacitance. In fact, there have been already two major perspectives to this phenomenon; one is to interpret this behavior as the manifestation of the intrinsic negative capacitance of the Landau barrier as originally suggested by Khan’s paper [*Nat. Mater.* 2015, 14, 182]. The other is to treat them as an artefact of complex nonlinear system involving the classical switching kinetics of the ferroelectrics [*J. Appl. Phys.* 2018, 123, 105102]. (Also, in this regard, we highly recommend the authors to refer to another critical study from Kim et al. [*Nano Lett.*, 2017, 17, 7796] which is not cited in this manuscript) We believe that the authors’ view is more related to the latter perspective based on their statement that the polarization flip is likely to be involved in the SS improvement, and that the continuous change of polarization is very improbable. Therefore, in the discussion, the section IV of this manuscript, we recommend the authors to clarify that there have been two different perspectives to this behavior, and the results of this experiment also strongly support the latter interpretation.

Response to comments:

Thanks for providing us a great comment about the works so far published. We have read these papers except Kim’s paper in *Nano. Lett.* These have discussed transient NC basically. In order to directly compare with SS in FET, we have been particular about so-called “quasi-static NC” rather than transient NC. On basis of this, we have considered the NC from the viewpoint of classical polarization switching in DC mode rather than transient AC mode. The fact that a huge amount of bound charge movement should be associated with the polarization switching is a main point in the present work obtained experimentally. Kim’s paper is very interesting and discussing a similar view in the S-T type AC analysis in addition to associating the NC with a specific polarization kinetics of KAI model. Although the physics behind with regard to the

bound charge movement is similar to the present work, our model does not necessarily require a specific domain switching kinetics in terms of whether it is like KAI, NLS or statistical-KAI or -NLS models, because the DC-mode polarization switching rather than high-speed switching has been focused in this paper as mentioned above.

Nevertheless, we would like to thank the reviewer#1 for useful comments which have made clear of our standpoints. We have added Kim's paper (Ref. 24) and another paper about ferroelectric switching kinetics (Ref. 29, *Scott J. F., Ferroelectrics, 503, 117 (2016)*) as the references. We also modified the introduction part as well as the section IV in the manuscript to state above standpoints more clearly. The modified text is as below:

Introduction part

...

However, NC effects are still under intensive debate¹⁶⁻²⁷. In fact, several arguments have been carried out that the NC region of FE material is intrinsically unstable or even impossible^{16,17}. Alternative explanations for the capacitance enhancement and transient NC effects in FE/PE stacks have been also proposed¹⁸⁻²⁴. For example, a feasibility of capacitance enhancement has been considered by a strong coupling between FE and PE layers¹⁸⁻²⁰, while the transient NC has been discussed from the viewpoint of classical switching kinetics of ferroelectrics²²⁻²⁴. In addition, the SS improvements in FETs so far reported suffer from critical problems such as large hysteresis observed, high voltage needed and operation frequency limited in actual experiments²⁵⁻²⁷.

Thus, it is urgently demanded to elucidate whether the quasi-static NC initially proposed can be really stabilized or not, and whether steep SS characteristics so far demonstrated is really promising for low-power CMOS applications or not. To provide useful insights to these critical questions, very careful examinations of FE/PE stacks should be carried out. A direct way to examine the NC effect in FE/PE stack is to investigate the voltage at the internal node, V_{int} , between FE and PE layers in DC mode, which makes possible of the direct comparison of V_{int} with SS. In this work, DC measurements are focused, and the stepwise V_{int} jumps at the coercive voltage, V_C , of FE layer in FE/PE stack are reported.

...

Section IV

Finally, the intrinsic difference between our view and the original proposal of NC effects in FE-based FETs is discussed. The V_{int} gain is actually obtained in both cases. In the present view, it is associated with the domain flip, on the assumption that an intermediate state between two polarized states in each domain is not stable quasi-statically (**Fig. 5(a)**). This requires that the gate voltage should be high enough to enable $V_F > V_C$. While in the original NC proposal, the intermediate state with a small polarization value can be stabilized with a small C_p , and the V_{int} gain is obtainable even at $V_F=0$ (**Fig. 5(b)**). It is still not perfectly clarified whether it is impossible to access to the intermediate polarized state quasi-statically or not. It is, however, strongly inferred that the continuous change of polarization in a domain is very improbable because of no experimental indication of V_F change between $-V_C$ and $+V_C$. It is also consistent

with the fact that no observation of the SS improvement under a small voltage swing has been reported so far.

Moreover, it is worthy of mentioning that the polarization reversal depends on specific domain switching kinetics such as Kolmogorov-Avrami-Ishibashi or nucleation-limited switching models in the multi-domain system^{24,29}. But this is beyond our consideration, because the DC-mode polarization switching rather than high-speed switching has been focused in this paper. When the polarization switching speed is concerned in device applications, the switching kinetics should be taken into account by including Landau switching and their modified versions as well.

2. The electrostatic model proposed by the authors in the section III seems to have an inaccurate expression or inadequately reflect the physics of the domain flips. First of all, we believe that the equation (4) and (5) seems wrong, or at least misleading. Starting from the total charge density $Q = Q_{free} + Q_{bound} = C_F V_F + P$ (i.e. eqn. (3)) and the Kirchhoff's voltage law, each voltage, V_{int} and V_F should be obtained as follows,

$$V_{int} = \frac{C_F}{C_p + C_F} V + \frac{P}{C_p + C_F}$$

$$V_F = \frac{C_p}{C_p + C_F} V - \frac{P}{C_p + C_F}$$

Therefore, the sign of the second term in the equations should be reversed. This inaccurate expression leads to unnecessary confusion associated with equation (7) because the sign of the second term in this equation become reversed with respect to equation (4).

More importantly, the equation (7) seems to be wrong in the first place. The internal voltage boosting factor, $\frac{\delta V_{int}}{\delta V}$, can be directly obtained by differentiating V_{int} the equation (4) (with the correct sign) with respect to V as follows,

$$\frac{\delta V_{int}}{\delta V} = \frac{C_F}{C_p + C_F} + \frac{1}{C_p + C_F} \frac{\delta P}{\delta V}$$

However, the equation (7) is not correct since the last term does not involve the derivative of the bound charge P with respect to the applied voltage V , but just the switching amount of the polarization ($2P$). We believe this wrong expression of the equation (7) results in a serious inconsistency with the actual physics. Unlike the explanation from this model, the voltage boosting is attributed to the fast switching rate ($\frac{\delta P}{\delta V}$) near (or above) V_C . As explained by Saha's model [*J. Appl. Phys.* 2018, 123, 105102], the overshooting voltage above V_C without the domain switching leads to the abrupt voltage drop ($\frac{\delta V_F}{\delta V} < 0$) once the retarded switching is initiated. However, the rate of the switching $\frac{\delta P}{\delta V}$ is obviously highly nonlinear function of ferroelectric voltage, which is abruptly increased near V_C . Therefore, since the amount of the

voltage drop is not determined by the magnitude of P but the switching rate $\frac{\delta P}{\delta V}$, it is highly improbable that the transient NC is maintained near $V \sim 0$ (far below the coercive voltage) where $\frac{\delta P}{\delta V}$ is very small. This is why the transient negative capacitance in the ferroelectric capacitor emerges near the coercive voltage. Although it seems reasonable that the stepwise voltage drop occurs due to the successive domain flips, it should be noted that even the zig-zag voltage drop after the biggest overshoot voltage is mainly concentrated near the coercive voltage (~ 0.6 V and ~ 1 V) of the ferroelectric as shown in Fig. 1(e).

Response to comments:

We would like to sincerely thank reviewer#1 for careful checking and comments, and to apologize our stupid mistakes.

In our experiments, positive polarization, P , is defined as in Fig. 4(a) and FE capacitor was pre-polarized by negative voltage in prior to measurements, as shown in Fig. (4b). Therefore, as the reviewer#1 made a comment, eq. (3) should be

$$Q = Q_{free} + Q_{bound} = C_F V_F - P, \quad (3)$$

rather than $Q = Q_{free} + Q_{bound} = C_F V_F + P$. Eq. (4) and (5) are as they are. And, as the reviewer pointed out, eq. (7) should be

$$\frac{\delta V_{int}}{\delta V} = \frac{C_F}{C_P + C_F} + \frac{\frac{\delta P}{\delta V}}{C_P + C_F} = \frac{C_F}{C_P + C_F} + \frac{\frac{2P_i}{\delta V}}{C_P + C_F} \quad (7),$$

if we assume a single domain polarization switching. Here, δV is an increment of V_g .

Thank you very much for pointing out careless mistakes. Since in our experiments, δV is mV, the 2nd term in eq. (7) becomes a huge. Eq. (8) should also be $\frac{2P_i}{C_F + C_P}$ in the multi-domain case, in which the internal voltage amplification can be also so large, if $2P_i$ is a fraction of the total P when compared with normal capacitive charges. It should occur near the coercive voltage, V_C , as the reviewer #1 mentioned. This is the main point obtained experimentally in the present work. The formulation is very important in a scientific paper, and we are really sorry about elementary mistakes. We have revised eq. (3), (7) and (8) together with explanation in the section III as:

It is here noted that two kinds of charges are involved in FE-CAP: bound charges (Q_{bound}) and free charges (Q_{free}), as shown in **Fig. 4(a)**, where the positive polarization, P , is defined as the arrow from left to right. Since FE-CAP is pre-polarized by a negative voltage in the experiments, $Q_{bound} = -P$ initially as shown in stage-1 in **Fig. 4(b)**. According to electrostatics, the relevant system should satisfy the following equations before the polarization switching:

$$V_F + V_{int} = V, \quad (1)$$

$$V_{int} = \frac{Q}{C_P}, \quad (2)$$

$$Q = Q_{free} + Q_{bound} = C_F V_F - P, \quad (3)$$

in which Q denotes total charges accumulated on each CAP. Assuming C_F is constant, V_{int} and V_F are expressed by

$$V_{int} = \frac{C_F}{C_F + C_p} V - \frac{P}{C_F + C_p}, \quad (4)$$

$$V_F = \frac{C_p}{C_F + C_p} V + \frac{P}{C_F + C_p}. \quad (5)$$

Since the bound charges are fixed in the case without polarization flipping, the FE/PE stack is similar to a PE/PE one. Thus,

$$\frac{\delta V_{int}}{\delta V} = \frac{C_F}{C_p + C_F} < 1. \quad (6)$$

On the other hand, when the polarization flipping occurs at $V_F \approx V_C$ (taking the forward sweeping as an example), the polarization, P , should change the direction in Eq (3), (4) and (5). Namely, a bunch of charges are transferred from FE-CAP to PE-one. Assuming FE layer is with a single domain,

$$\frac{\delta V_{int}}{\delta V} = \frac{C_F}{C_p + C_F} + \frac{\frac{2P}{\delta V}}{C_p + C_F}. \quad (7)$$

...

And, the stepwise V_{int} jumps is analytically obtained as

$$\frac{\delta V_{int}}{\delta V} = \frac{C_F}{C_F + C_p} + \frac{\frac{2P_i}{\delta V}}{C_F + C_p}, \quad (8)$$

in which P_i ($i=1, 2, 3, \dots$) is the polarization of each domain in FE film.

And, yes, Saha's paper which we have cited explains the $\frac{\delta V_F}{\delta V} < 0$ from the viewpoint of voltage overshooting which may occur transiently. Our experiments, however, was performed in DC mode and no voltage overshooting was expected in this case. Therefore, we believe the $\frac{\delta V_F}{\delta V} < 0$ is associated with depolarization feedback from PE-CAP in polarization switching at $\pm V_C$. Since our model is based on the assumption that the intermediate state of polarization cannot be stabilized. This means, QS-NC at $V_F=0$ proposed in the initial theory of NC is highly improbable based on our model, as the reviewer pointed out.

Concerning the zig-zag voltage drop after the biggest, yes, as the reviewer commented, it should

also occur at V_C if we assume V_C for each domain is identical and there is no domain-domain interaction in successive switching in an ideal case. In reality, there should be a distribution of V_C in the multiple-domain case. Also, the domain-domain interaction may be induced and the internal field inside the film may be reduced in the first domain switching. As a matter of fact, the stepwise voltage drops of V_F after the biggest one are a little bit smaller than V_C . We have revised some sentences in the section III to state these points more clearly as:

In actual FE films, multiple domains are involved and not all domains flip simultaneously. Here, we assume that each domain flips in FE film independently and V_C for each domain flipping is identical. When a certain domain flips, V_F should be reduced due to a finite amount of charge transfer from FE-CAP to PE-one. It means that remaining domains cannot flip before further increase in V_F to V_C . When V_F is further increased to V_C , another domain flips, followed by V_F drop again.

.....

In addition, it should be mentioned that V_C in each domain is not identical but has a distribution in a FE layer, and that the domain-domain interaction may be induced and the internal field inside the film may be reduced in the first domain switching. These effects should affect the voltages for successive domain flipping. In fact, our results in Fig. 1(d) show that the successive domain flips, following the biggest one, occur at V_F values slightly smaller than V_C .

3. We believe that the hysteresis engineering scheme proposed by the authors in the section 3 of the supplementary manuscript is also not free from the inconsistency of their models we addressed above. As a matter of fact, the modulation the capacitance of the external serial capacitor has been experimentally tested by Khan et al. [Appl. Phys. Lett. 2017, 111, 253501] Their results (Fig.3 of the Khan's paper) showed that the decrease of the external capacitor does not necessarily lead to the hysteresis free (quasi-static) operation of the ferroelectric capacitor. As shown in Fig.3, a less steep load line of a smaller capacitance leads to a decrease of the amount of the domain reversal, not a decrease of the magnitude of the hysteresis. This is because the transient voltage drop only emerges where $\frac{\delta P}{\delta V}$ is large enough, which is near the coercive voltage.

Response to comments:

Thanks for a critical comment. Concerning the hysteresis discussion, we have considered the behaviors as follows.

In the single domain case, both amplitude and position of V_{int} jump are controlled by PE capacitance (C_P) because the intermediate state of P cannot be stabilized in our model. It means that decrease of the amount of the domain reversal is impossible. In this case, hysteresis is determined by the position of V_{int} gain in the forward and backward V sweepings in FE/PE stacked capacitor case. As a result, nearly hysteresis-free case can be made possible by controlling C_P . The hysteresis-free in the multi-domain case, on the other hand, is very challenging but more important practically. In this case, the switching does not occur in all domains simultaneously. As mentioned above, when V_F drops with a specific domain switching,

remaining domains will not be switched before further increase of V_F to V_C . Resultantly, a zigzag V_{int} jump and Q - V_F curve with a hysteresis should be obtained. As the reviewer pointed out, in a case with small C_P , a decrease of the amount of the domain reversal may occur. Meanwhile, if both P_i and C_P could be controlled accurately, it is possible to achieve a hysteresis-free path in the zigzag V_{int} - V and Q - V_F curve, as show in Fig. 4(c) and Fig. S4(c). This is not perfectly hysteresis-free, but it might be helpful for increasing/decreasing I_D in the subthreshold region in a NCFET-like transistor. As a matter of fact, it is not so easy to control the hysteresis to achieve the case in Fig. 4(c). Due to the bias-dependence of semiconductor capacitance, the internal potential control in FET with FE/PE gate stack is rather different from that in FE/PE stacked capacitor. Furthermore, we cannot control materials properties perfectly such as domain size and the number of polarization charges. In addition, the coercive field for each domain flipping is not identical. So, the hysteresis is mostly reported in our and published experiments. Nevertheless, a single domain ferroelectric film might be possible in aggressively scaled FETs. This is our naive and optimistic hope.

We would like to sincerely thank to the reviewer#1's careful comments, and we have modified Section III in the manuscript and section IV in the supplementary material to state our above consideration more clearly. The modified parts are as below.

Section III in the manuscript:

Hysteresis of the V_{int} gain is critical for steep SS FETs. Based on the present understanding, nearly hysteresis-free V_{int} gain is made possible in principle by inserting an appropriate C_P in the case of the single domain FE in FE/PE stack. The requirement for this condition is the same as that for stabilizing NC effects originally proposed. In case of actual FE layer containing multiple domains, the hysteresis-free V_{int} gain is very difficult to achieve. Here, the biggest V_{int} gain is focused. Since the V value and amplitude of V_{int} jump are controlled by C_P and P_1 , respectively, it becomes possible that the V_{int} jump in the forward V sweeping may overlap in the main region with that in the backward one by adjusting a large P_1 under an optimum C_P , (**Fig. 4(c)**). Resultantly, the subthreshold hysteresis in FETs might be much reduced in the overlapped range. This is practically very challenging when the polarization of the biggest domain in FE films and the bias dependence of semiconductor capacitance are considered. The detailed discussion of hysteresis is shown in supplementary materials IV.

Section IV in the supplementary material:

The hysteresis control in the multiple-domain case is very challenging but practically of a great importance. To find out how to control the subthreshold hysteresis in FET with FE gate stack, the biggest V_{int} gain is focused. Since C_P and P_1 dominantly affect the position and the magnitude of V_{int} gain, the hysteresis in FET might be minimized by achieving an overlapping region of V_{int} gain in both forward and backward sweepings, as schematically shown in **Fig. 4S(c)**. This will become possible in the following conditions.

$$C_P \approx \frac{\sum_{i=1}^{i=n} P_i - C_F V_C}{V_C}, \quad (\text{S6})$$

$$P_1 > \frac{1}{2} \sum_{i=1}^{i=n} P_i. \quad (S7)$$

Here, n is the number of domains. The requirement for C_p looks similar to that in the single domain case, while that for P_1 is complicated, because the regions showing V_{int} gains in forward and backward sweepings may be overlapped only in the condition of Eq. S(7). Nevertheless, the total sum of polarizations over the successive switching is apparently equivalent to the stabilized intermediate polarization state assumed in the original single domain NC theory. The key difference of the present view from the original NC theory is that the intermediate state of each domain is not stable. As a matter of fact, it is not easy but rather impossible to control the polarization and switching of each domain. In addition, the coercive field in each domain is not always identical. Thus, nearly hysteresis-free FETs will be made possible only in a single domain FE case, though the state on the switching process cannot be stabilized.

We would sincerely appreciate the reviewer #1 for careful review and helpful comments.

Reviewer #2

This work is dedicated to studying and modeling internal potential jumps like a stepwise near the coercive voltage. The authors A relationship of this effect with the steep SS in FET is also demonstrated experimentally by connecting a FE capacitor to a simple metal-oxide-semiconductor FET.

1) This zig-zag effect in polarization is not new at all and overall, it is a too focused technical work to be publishable in Nature Communications. To be very clear on the novelty, the experimental proof of multiple jumps due to multiple ferroelectric domains has been previously reported by A. Saeidi et al, 'Negative capacitance as digital and analog performance booster for complementary mos transistors,' Publication date 2018/4/25, Journal arXiv preprint arXiv:1804.09622 (now published in Scientific Reports), and also showed in various conference talks. The authors seem not to be aware or they just skip such obvious literature available for polycrystalline PZT used in similar experiments, which is not acceptable.

Response to comments:

First of all, we would like to thank the reviewer#2 for critical comments. Although the reviewer#2 made a comment that our paper is too technical to be published in NC, we think a bit differently. When we submitted this paper, we did not know the arXiv preprint the reviewer#2 commented. We specially cared about experimental method in the internal potential measurement in DC mode in terms of the impedance in the signal sampling system. The point was not precisely described in the main text. Therefore, we have revised the section I by including this point in the text rather than in the supplement. With this carefully designed DC measurement, our results showed the internal potential jumps at $\pm V_C$ of FE layer. Now we have understood the zig-zag effect is also shown in Scientific Report. However, an importance of the internal voltage measurement and the internal potential jumps corresponding to $\pm V_C$ of FE film have not been pointed out in this paper. We believe that well-defined experimental set-ups and systems are the most important for experimentalists to understand the results physically.

We have revised section I in our manuscript to state our standpoints more clearly as below:

To correctly estimate V_F in FE/PE stack in DC mode, a PE/PE stack was firstly tested quantitatively. In an ideal case, total charges at the internal metal, which is electrically floating, is conserved as long as no leakage current through both CAPs is maintained (**Fig. 1(a)**). However, since the DC output impedance at the floating node is infinite in the case of no

leakage current through capacitors, it is substantially impossible to estimate V_{int} experimentally. In actual cases, a finite resistance of capacitors enables the V_{int} measurement in DC-mode, while a time constant should be considered. More importantly, a small amount of charge-flow from the internal metal to a measurement system should also be paid attention to, because the measured voltage in itself is significantly affected by the input impedance in measurement system. The equivalent circuit concerned in this work is shown in **Fig. 1(b)**. Based on the formulation shown in supplementary materials I²⁸ and the measured resistance of PE CAPs ($\sim 10^{12} \Omega$), a voltage measurement system with the input impedance higher than $\sim 10^{14} \Omega$ was needed for accurate DC-mode measurement. We used a high input impedance voltmeter equipped with a high precision current preamplifier, which adjusted a voltage to suppress a current flow down to zero (sub-fA level). By doing so, the effective input impedance was enhanced up to $10^{16} \Omega$. **Fig. 1(c)** compares V_{int} measured in PE/PE stack in two cases with the present system and conventional voltmeter with an input impedance of $\sim 10^{10} \Omega$. When the present system was employed, the accurate measurement of V_{int} was successfully assured for capacitors with down to 1pF. Namely, the measurement system does not affect voltage measurement at all.

Fig. 1 (a) Ideal and (b) actual equivalent circuits in the internal potential measurement in PE/PE stack capacitor. (c) V_{int} , with $V=1V$ and $C_1=200$ pF, as a function of C_2 from 1 pF to 33 nF for two cases using high-Z and normal-Z DC measurement systems. Only high-Z system enables to estimate the internal potential in DC mode measurement.

We also added Figure S3 to show that V_F jump positions are consistently near $\pm V_C$ even by changing C_P from 0.1 pF to 15 nF as below.

Fig. S3 shows V_F - V characteristics in FE/PE stacks with four kinds of C_P . In all cases, the first

and biggest V_F drop, corresponding to the first and biggest V_{int} jump, occurs near $\pm V_C$ of FE film

Fig. S3 V_F - V characteristics in FE/PE stack capacitors with four kinds of C_P .

2) Even if the modeling part seems to have some originality, this is not enough for publication in Nature Communications, therefore it should be submitted to a more technical journal. However, even in the modeling part the authors speculate that every domain flip could enable an internal potential gain in FE/PE stack. The model associated with the domain flip, on the assumption that an intermediate state between two polarized states of each domain is not stable, is put in contrast with the NC original effect (Fig. 4). Apart the multi-domain interactions that are correctly addressed by the authors, even the full model is not new, as it is known that many authors wrongly reported steep-slopes in transistors due to polarization switching and not due to true NC effect. Therefore, the report made and the results are not really surprising for an expert and are not of practical use in low power transistors/circuits, as the authors themselves are recognizing in the conclusion section.

Response to comments:

Concerning the zig-zag model, key point is that charge redistribution may change the potential profile in stacked system and the following switching is not probable in principle. In reality, there should be E_C distribution, and there might be the case with a seamless switching. But the zig-zag

should be more general behavior. Yes, there are a huge amount of papers making comments that NC is not actually usable for low power applications. However, most of them are based on naive views on ferroelectric properties and their understandings. Therefore, direct experimental evidences on the basis of well-defined experimental systems, followed by physical understanding on the basis of experimental results, have been strongly needed to clarify these comments under debates.

The polarization switching issues have been investigated since ferroelectric films were found. It is interesting to see the Merz's paper published in 1954, in which the importance of polarization switching mechanism was mentioned as the same level as the currently published papers. This fact means that it is not so simple to conclude that NC effect is not usable. Furthermore, we have to attend that the definition of NC effects depends on each personality. Someone defines NC effect by the internal potential amplification, while others only admit the model originally proposed by the Landau model for the ferroelectricity. And recently many discussions are moving to the transient NC effects without taking account of experimental conditions and taking care of many time constants. Therefore, we do not want to discuss about which is true NC or apparent NC. We should discuss about it on the physics basis step-by-step. We are not saying that other ferroelectric films should show the same behavior, because the switching mechanism also depends on the ferroelectric materials and sizes. We would be happy if you could understand our standpoints.

3) The experiment itself provides useful data but it is not original and, as setup, was part of many publications in the past, many not properly cited by the authors. The authors are not even carefully considering or discussing the parasitics related to such experiment.

In conclusion, I cannot see how this work can be published in Nature Communications and should be redirected to another IEEE journal.

Response to comments:

Yes, the internal potential measurements in FE/PE stacks are not new but have been tried. But the accuracy of experimental results is not necessarily reliable, because no special cares have been taken about the experimental systems. We believe the internal potential measurement in FE/PE stack has been carried out for the first time in carefully designed DC measurement system. In our measurement, we deeply considered about the effect of impedance in the measurement system as mentioned above. This is a key point for accurate measurement of internal potential in high impedance system. If this is the “parasitics” the reviewer mentioned, we have considered it very much. In addition, in the revised manuscript, we have also added further consideration about the effect of leakage through FE-CAP as below.

Resultantly, the equivalent circuit of the V_{int} measurement in FE/PE system shown in **Fig. 2(a)** was assumed, because impedances in both PE-CAP and measurement system were much higher than that in PZT.

Fig. 2(a) Equivalent circuit of internal potential, V_{int} , measurement in FE/PE system. Note that high impedance system is needed to get the accurate V_{int} .

.....

It is worthy of mentioning that the measured voltage was time-dependent and it gradually increased due to a finite resistance in FE layer as mentioned above. Therefore, the absolute value of V_{int} is not as expected in C_{PE} - C_{FE} circuit. Both V and V_{int} are, however, very correct and V_F obtained by $V - V_{int}$ should be correct quantitatively. This is further confirmed by the results in **Fig. S3** where V_F positions corresponding to the biggest V_{int} jumps are consistently near $\pm V_C$ by changing C_P from 0.1 nF to 15 nF.

We are very sure it will be very useful for experimentalists working not only for NC effects but also for ferroelectric generals. And many thanks for careful review.

Reviewer #3

This paper advances the understanding of the so-called “negative capacitance” phenomena that have attracted a great deal of interest in the last few years. The authors treat the multi-domain ferroelectric thin film under discussion as a combination of small Landau domains, with each domain following the Landau theory of ferroelectricity. This approach enables the authors to successfully explain those sub-60 SS data often reported for Id-Vg measurements resulting from switching transients. Instead of adopting the popular “quasi-static negative capacitance” argument, the authors show convincing experimental data and clearly state that the apparent “negative capacitance” effect only occurs during ferroelectric switching for the multi-domain FE, which requires $V_g > V_c$. This is very important contribution that deserves quick publication to educate the readers who may very well have been exposed to papers in the literature that erroneously claim the observation of “quasi-static negative capacitance”.

However, there are some minor flaws in the description of the experimental setups and the details of the measurements, which need clarification before this paper is published.

1. The equivalent circuits drawn in Fig. 1(a) and Fig. 2(a) give the impression that the M/PZT/M sample being measured is represented by a capacitance without any leakage current. In reality, any realistic M/PZT/M sample has a finite leakage current, which should be represented by a parallel resistance (which is voltage dependent) across the M/PZT/M sample. Actually, in the “Supplementary Material” Section, Fig.S1(b) depicts the more appropriate equivalent circuit, which should be used in the main manuscript.

3. In Fig. 1 (b), when the applied voltage is +/-6V, the internal voltage is around +/- 5V. Since according to the manuscript, CFE is 0.25nF and CP is 0.5nF, the measurement results indicate that the voltage divider is not based on the capacitances but rather the impedances of the two capacitors, which is exactly why the previous comment #1 was made by the reviewer. The authors need to clarify this. In connection with this, the authors should explain in detail how the DC measurement of Fig. 1 (b) is performed.

Response to comments:

Thanks for a critical comment on the measurement. Yes, these are important points in our experiments. For simplicity, we described the simple circuits in the main text, and more realistic one in the supplement material in the initial version. We have revised the section I significantly by adding/changing figures, talking about the details how the DC measurement was designed and carried out, and discussing the leakage effect on the measurement results.

a) Firstly, the careful designing of DC measurement system is the key point in our study. We have most carefully paid attention to the measurement input impedance which has been taken less care of so far. When it is very high, the measured voltage should be correct. If this voltage is right, the voltage applied on FE at a moment is right. This means we have two kinds of voltages (V and V_{int}) very accurate at a moment. These have been experimentally confirmed in well-defined PE/PE stacks. We have modified the first paragraph in section *I* by incorporating Fig. S1 to the main text and stated above points more clearly as below.

To correctly estimate V_F in FE/PE stack in DC mode, a PE/PE stack was firstly tested quantitatively. In an ideal case, total charges at the internal metal, which is electrically floating, is conserved as long as no leakage current through both CAPs is maintained (**Fig. 1(a)**). However, since the DC output impedance at the floating node is infinite in the case of no leakage current through capacitors, it is substantially impossible to estimate V_{int} experimentally. In actual cases, a finite resistance of capacitors enables the V_{int} measurement in DC-mode, while a time constant should be considered. More importantly, a small amount of charge-flow from the internal metal to a measurement system should also be paid attention to, because the measured voltage in itself is significantly affected by the input impedance in measurement system. The equivalent circuit concerned in this work is shown in **Fig. 1(b)**. Based on the formulation results in supplementary materials *I*²⁸ and the measured resistance of PE CAPs ($\sim 10^{12} \Omega$), a voltage measurement system with the input impedance higher than $\sim 10^{14} \Omega$ was needed for accurate DC-mode measurement. We used a high input impedance voltmeter equipped with a high precision current preamplifier, which adjusts a voltage to suppress a current flow down to zero (sub-fA level). By doing so, the effective input impedance was enhanced up to $10^{16} \Omega$. **Fig. 1(c)** compares V_{int} measured in PE/PE stack in two cases with the present system and conventional voltmeter with an input impedance of $\sim 10^{10} \Omega$. When the present system was employed, the accurate measurement of V_{int} was successfully assured for capacitors with down to 1pF. Namely, it has experimentally been elucidated that the measurement system does not affect measured results at all.

Fig. 1 (a) Ideal and (b) actual equivalent circuits in the internal potential measurement in PE/PE stack capacitor. (c) V_{int} , with $V=1V$ and $C_1=200$ pF, as a function of C_2 from 1 pF to 33 nF for two cases using high-Z and normal-Z DC measurement systems. Only high-Z system enables to estimate the internal potential in DC mode measurement.

b) Concerning the actual measurement of FE/PE stacks, we selected low leakage PZT and conventional capacitors in our work, but it is true that both have leakage currents. We measured DC $I-V$ ($<1V$) characteristics. Roughly, PZT has ~ 50 G Ω in a low bias region and ceramic capacitor has $\sim 1T\Omega$ as mentioned above. Therefore, we have ignored the leakage through the PE-CAP as well as the V_{int} measurement system with high input impedance, and obtained an analytical formula from (S3).

$$V_{int} \approx V \left(1 - \frac{C_2}{C_1 + C_2} \exp\left(-\frac{t}{R_1(C_1 + C_2)}\right) \right)$$

$$\tau = R_1(C_1 + C_2) \sim 50G\Omega \cdot 100 \text{ pF} \sim 5 \text{ s}$$

V_{int} observed experimentally increased with a time in a range of several of seconds, and we took the data in \sim a second. Furthermore, As the reviewer#3 pointed out, the resistance of PZT is not a constant but depends on bias. Thus, the absolute value of measured voltage is not as expected from $C_{FE}-C_{PE}$ circuit. The voltage division looks determined by the impedance division rather than capacitance one. However, both V and V_{int} are very correct and V_F obtained by $V-V_{int}$ should be correct quantitatively. This is the point in our experiments. When the input impedance of measurement system is very high, the measured voltage should be correct, even though a little bit leaky samples are measured. If this voltage is right, the voltage applied on FE at a moment is right. This is a key concept. Thus, it is very sure that the voltage jumping-up (or down) was clearly detected because V_F corresponding to the first domain switching is very near to $\pm V_C$ even by changing the value of C_p . However, in most of experiments, the careful measurement set-up has not been taken care. We have revised the second paragraph and Fig. 2(a) (Fig. 1(a) in the initial version) in section I to describe how the measurement was performed and to state above points clearly as below.

In actual measurements of V_{int} in FE/PE stack, commercially available PZT films with Pt electrodes were used as the FE-CAP. The typical charge-voltage ($Q-V_F$), capacitance-voltage ($C-V_F$) and leakage characteristics are shown in supplementary materials II. The capacitance at the center of $C-V_F$ characteristics was ~ 0.25 nF and the resistance at 1 V was $\sim 5 \times 10^{10}$ Ω . Resultantly, the equivalent circuit of the V_{int} measurement in FE/PE system shown in Fig. 2(a) was assumed, because impedances in both PE-CAP and measurement system were much higher than that in PZT. In prior to measurements, the FE capacitor was polarized by a negative voltage, and the internal terminal of FE/PE stack was grounded to remove the unknown charges stored. More details in measurements are described in the method part.

...

It is worthy of mentioning that the measured voltage was time-dependent and it gradually increased due to a finite resistance in FE layer as mentioned above. Therefore, the absolute value of V_{int} is not as expected in $C_{PE}-C_{FE}$ circuit. Both V and V_{int} are, however, very correct and V_F obtained by $V-V_{int}$ should be correct quantitatively. This is further confirmed by the results in Fig. S3 where V_F positions corresponding to the biggest V_{int} jumps are consistently near $\pm V_C$ by changing C_P from 0.1 nF to 15 nF.

Fig. 2(a) Equivalent circuit of internal potential, V_{int} , measurement in FE/PE system. Note that high impedance system is needed to get the accurate V_{int} .

And, we have added a figure in the Section III of supplementary material as:

Fig. S3 V_F - V characteristics in FE/PE stack capacitors with four kinds of C_p .

c) For the PE/MOSFET, we have confirmed that the leakage in MOSFET was 1 pA under 1V. It was almost negligible comparing with that in FE layer as in FE/PE case. Therefore, as the reviewer kindly suggested, we have revised Fig. 3(a) (Fig. 2(a) in the initial version) as below.

Fig. 3(a) Equivalent circuit for measuring I_{DS} - V_{GS} and V_{int} - V_{GS} curves in FE/MOSFET system. Note that V_{int} - V_{GS} and I_{DS} - V_{GS} were measured separately for the same system and under the same V_{DS} .

d) Moreover, we have revised the method part by adding the details of the measurement as below.

Before measurement, each terminal of FE/PE and FE/MOSFET circuit was grounded to remove unknown charges left inside. Then the internal potential, V_{int} , were measured by sweeping total voltage, V , with a step of 50 mV. In each step, a wait time of 0.1s was set for stabilizing the voltage and a total time of 0.5s was taken.

2. For the measurement setup, the authors emphasize the importance of the ‘high-Z’ voltage meter, and set the input impedance of the voltage meter to be 10^{16} ohms. The authors should explain in detail how this is accomplished.

Response to comments:

Yes, this was a key in the measurement. Otherwise, we have to take account of input impedance value for analyzing the decay time in the above equation. We carried out experiments as follows.

As we have discussed above, C_p was roughly 10^{12} Ω , so we needed the input impedance larger than 10^{14} Ω for the voltage measurement system for keeping 1% accuracy. As known well, it is not easy to access to such high impedance voltmeter, which is generally $10^{10}\sim 10^{12}$ Ω . Thus, we used a high impedance voltmeter equipped with a high precision current preamplifier, which indicates a voltage to suppress a measurement current down to zero (sub-fA level). By doing so, effective input impedance can be enhanced up to 10^{16} Ω if we believe the specification sheet. In fact, as shown in Fig. 1(c), it was experimentally confirmed that this measurement was actually quite effective for the voltage measurement in the high output impedance system. As mentioned above, we have revised section I in our revised manuscript to state these points more clearly as:

Based on the formulation results in supplementary materials I^{28} and the measured resistance of PE CAPs ($\sim 10^{12}$ Ω), a voltage measurement system with the input impedance higher than $\sim 10^{14}$ Ω was needed for accurate DC-mode measurement. We used a high input impedance voltmeter equipped with a high precision current preamplifier, which adjusts a voltage to suppress a current flow down to zero (sub-fA level). By doing so, the effective input impedance was enhanced up to 10^{16} Ω . Fig. 1(c) compares V_{int} measured in PE/PE stack in two cases with the present system and conventional voltmeter with an input impedance of $\sim 10^{10}$ Ω . When the present system was employed, the accurate measurement of V_{int} was successfully assured for capacitors with down to 1pF. Namely, it has experimentally been elucidated that the measurement system does not affect measured results at all.

4. In Fig. 3, the authors assume that all the small domains have the same coercive field, while in reality, the ferroelectric domains may have different orientations in a given sample. Since the applied voltage induces an electric field (E_{app}) perpendicular to the electrodes, some domains may require $E_{\text{app}} > E_c$ to switch due to the domain mis-alignment. Therefore, it is more realistic to assume that the effective coercive field has a distribution instead of one identical value. The authors are urged to take this into consideration in their revision.

Response to comments:

Thank you for a great comment. Yes, we totally agree with the reviewer#3. The domain switching kinetics in ferroelectric films is still under debate in the ferroelectric research community. We would not like to discuss about the polarization switching kinetics such as KAI, NLS or statistical-KAI or -NLS in this paper. In reality, there is a distribution of E_C , as the reviewer#3 commented. It may lead to a statistical variation in the internal potential jumping. Furthermore, this effect should be dependent on ferroelectric material and thickness in terms of switching kinetics. Therefore, we can also discuss the results from the viewpoints that the polarization switching kinetics and its statistics can affect the zig-zag oscillating characteristics of V_{int} . Thank you very much for the suggestions. We have revised the final paragraph in Section III to state these points more clearly.

In addition, it should be mentioned that V_C in each domain is not identical but has a distribution in a FE layer, and that the domain-domain interaction may be induced and the internal field inside the film may be reduced in the first domain switching. These effects should affect the voltage for successive domain flipping. In fact, our results in Fig. 1(d) show that the successive domain flips, following the biggest one, occur at V_F values slightly smaller than V_C .

5. Concerning Fig. 2, prior to the I_d - V_g measurement, the authors should state whether they discharged the ferroelectric capacitor and the internal node. The reason for this question is that when an external capacitor is connected to the gate, the threshold voltage may shift to the right (for NMOS). If the internal node has finite charge, then the threshold voltage may change differently. This charge may also affect the SS results. The authors state that in the FE-PE measurement, the internal node is discharged, but for the FE-MOS measurement, this is not mentioned. Please clarify this.

Response to comments:

Yes, we have always be careful about it, because our interpretation of experimental results are based on the charge redistribution at the internal node associated with the switching. The internal node was definitely grounded before each measurement including FE-MOS system, and PZT was poled in prior to measurement.

And, thanks for significantly important comments. Although this is very important, we missed it in the text. In the revised version, we have added a sentence as below:

It is mandatory to take care of the unknown charge conservation at the internal node in FE/MOSFET case like that in FE/PE one before measurement, because floating charges should affect the charge dynamics in the domain reversal. This is particularly critical for considering hysteresis in I - V characteristics of FE/MOSFET system.

–

We would sincerely appreciate the reviewer #3 for very positive comments and advises.

Review on the revised manuscript “Stepwise internal potential jumps caused by multiple-domain polarization flips in metal/ferroelectric/metal/paraelectric/metal stack” by Xiuyan Li, and Akira Toriumi.

(Ms# NCOMMS-19-10207A)

In this revised manuscript, the authors mainly revised the technical points raised by the reviewers, including the errors in the electrostatic equations, which certainly improved the work. Nevertheless, the technical context of this work is rather outdated given the circumstance that the following several critical works are now published.

On the quasi-static NC (e.g. I. Luk’yanchuk et al. [Phys. Rev. B 2018, 98, 024107], [Commun. Phys. 2019, 2, 1.]), and comprehensive review papers (J. Iniguez et al., Ferroelectric Negative Capacitance, *Nature Reviews*, 2019, and H. W. Park et al., Modeling of Negative Capacitance in Ferroelectric Thin Film, *Advanced Materials*, 2019)

It seems that the authors are not very well updated with the recent advances made in the field, which can give a quite nice understanding of the observed internal voltage boosting. The only difference is the wiggling response of the voltage response, which was ascribed to the abrupt and irregular domain switching. This is a correct interpretation, but such behavior in multi-domain ferroelectric material has been well known in the field even for the bulk crystal from early time – Barkhausen pulses (For example, Barkhausen Pulses in Barium Titanate, *Chynoweth. Phys. Rev.* **110**, 1316 (1958).

Overall, the observed internal voltage boosting could be understood in a very straightforward manner. When a sufficiently high voltage (to make $V_F > V_C$) is applied, a certain domain flips, and the corresponding amount of charge varies. This amount of charge is too large to be compensated by the PE layer, so the only method not to violate the charge conservation law is to increase the voltage across the PE layer over the applied voltage. This is especially the case near V_C . Then, the voltage across the FE layer must be dropped. This is all that the authors have observed from their test setup.

Besides, similar experiments have been performed by other authors (Jo et al., Negative Capacitance in Organic/Ferroelectric Capacitor to Implement Steep Switching MOS Devices, *Nano Lett.* 2015, 15, 4553–4556) although the authors of that work have completely misinterpreted their results. The authors of the present manuscript used a better experimental setup and did not fall into the wrong conclusion. Nonetheless, the experimental observations are fundamentally identical to the other work.

Another critical conceptual problem of this work is as follows: the authors compared the MFM-MIM (or MIS, e.g., MFMIM or MFMIS) with MFIM (or MFIS) systems. These two systems are completely different regarding the boundary conditions. In the former case, the intervening metal layer provides a sufficient compensating charge in response to the change in the ferroelectric bound charge. Also, any (local) change in the FE layer, such as local domain reversal, is responded by the entire PE layer due to the presence of the intervening metal layer, which shorts out the surfaces of the FE and PE layers. In contrast, the latter case has a completely different behavior; the FE and PE layers must respond to the external stimuli together as a single body since there is no intervening metal layer that separates the two. Please refer to Hoffmann et al. [*Nanoscale* 2018, 10, 10891]. Therefore, the direct comparison made in Fig. 4 does not make any sense.

Moreover, the result from the load line analysis of MFMIM structure in Fig. S4(b) would not be directly applied to the NCFET operation. Since a semiconductor has a highly nonlinear capacitance depending on the voltage region, the hysteresis engineering by modulating the external capacitance would be highly complicated than this simple scheme presented in Fig. S4.

While the authors’ experimental setup to exclude the charge exchange between the internal node and

ground through the V-meter, which make it possible to estimate the variation in the node voltage in a quasi-static manner, is plausible, the presented arguments do not contribute that much to the current understanding of the field.

Reviewer #2 (Remarks to the Author):

The authors have very carefully addressed all the comments and remarks. They have also improved the text of the article and the references.

Especially, they explained how they considered the effect of impedance in the measurement system for accurate measurement of internal potential in high impedance system. In the new version of revised manuscript, they usefully added further consideration about the effect of leakage through FE-CAP. All these clarifications are very useful and improve the trust in the result.

Overall, the paper offers a very good insight into the zig-zag modeling in polarization behavior in polycrystalline PZT capacitors and there is indeed a difference between the author's view taking into account domain flip and the original proposal of NC effects in FE-based FETs by Salahuddin. This is still debatable and to be investigated in different ferroelectric material systems and compare the various models.

In conclusion, while I still strongly believe this paper is very focused technically and does not have a high level of novelty (the zig-zag effect itself is not new and the novelty part is about modelling and discussion of this effect) it includes very useful data and discussions to advance the field of negative capacitance. Therefore, I agree with the publication, provided that the Editors considers this novelty on modelling enough.

Reviewer #3 (Remarks to the Author):

The reviewer is satisfied with the authors' response to comments 1, 2, 4 and 5. However, the reviewer has several further comments/questions concerning #3, as follows:

1. In Fig. S3, the curve corresponding to $C_p=1.5\text{nF}$ is identical to the curve in Fig. 2(d), which, as described in the text, is the result when $C_p=0.5\text{nF}$. This obvious inconsistency may be a small mistake, but the reviewer still hopes that the authors could correct this and provides an explanation.

2. In their response, the authors admitted that the leakage of the FE capacitor was not negligible, and thus in the non-switching region the measured V_{int} and V do not follow the CFE-CDE voltage divider rule. They calculated that the RC constant was roughly 5s. If this is indeed the case, then the reviewer has the following critical concerns.

(1). The author said that the data was taken after $\sim 1\text{s}$. Compared to the RC time constant of 5s, $\sim 1\text{s}$ is a relatively short time during the transient when the rate of change is large, and any variation of the time used to take the data could significantly affect the measured V_{int} .

(2). With this leakage path, the internal node is no longer isolated, and thus the net charge on the internal node can accumulate during the measurement.

(3) During polarization switching, both the capacitance and the resistance of the ferroelectric capacitor become nonlinear, and thus the RC time "constant" during polarization switching is not a constant. Also, due to the randomness of the polarization switching according to the nucleation-limit-switching model, the resistance and capacitance may have some variations during switching. Based on these concerns, it is reasonable for the reviewer to suspect that the experimental results are not sufficiently solid to support the conclusion of this article. The zigzag curve during polarization switching could be caused by the measurement itself, or the randomness of the RC time constant during switching, instead of the explanation based on the Landau theory.

3. The experimental results shown in Fig. S3 are mysterious. The reviewer compared the curves for $C_p=0.1\text{nF}$ and $C_p=15\text{nF}$, and made these observations

(1). Considering the $C_p=0.1\text{nF}$ case: In the forward sweeping non-switching region, V changes

from -5V to -2V, and the V_F changes from -1V to 1.5V, which means almost all the applied voltage drops across the FE. In this case, the RC time constant is $\sim 5s$

(2). Considering the $C_p=15nF$ case: In the forward sweeping non-switching region, V changes from -6V to 4V, and the V_F changes from -1V to 1.3V. This time, most of the applied voltage drops across the DE. The RC constant is $\sim 750s$.

In the non-switching region, based on the capacitance-voltage divider rule, a larger C_p gives rise to a higher V_F . Also, a larger RC time constant means that the measurement is less affected by the leakage, such that the results should be closer to the ideal capacitance- voltage divider case. But the results shown in the figure are clearly not consistent with the above analysis.

The reviewer will recommend publication if the authors can provide satisfactory response to the comments presented above.

Reviewers' comments:

Reviewer #1 (Remarks to the Author):

Review on the revised manuscript “Stepwise internal potential jumps caused by multiple-domain polarization flips in metal/ferroelectric/metal/paraelectric/metal stack” by Xiuyan Li, and Akira Toriumi. (Ms# NCOMMS-19-10207A)

In this revised manuscript, the authors mainly revised the technical points raised by the reviewers, including the errors in the electrostatic equations, which certainly improved the work. Nevertheless, the technical context of this work is rather outdated given the circumstance that the following several critical works are now published on the quasi-static NC (e.g. I. Luk'yanchuk et al. [Phys. Rev. B 2018, 98, 024107], [Commun. Phys. 2019, 2, 1.]), and comprehensive review papers (J. Iniguez et al., Ferroelectric Negative Capacitance, Nature Reviews, 2019, and H. W. Park et al., Modeling of Negative Capacitance in Ferroelectric Thin Film, Advanced Materials, 2019) It seems that the authors are not very well updated with the recent advances made in the field, which can give a quite nice understanding of the observed internal voltage boosting. The only difference is the wiggling response of the voltage response, which was ascribed to the abrupt and irregular domain switching. This is a correct interpretation, but such behavior in multi-domain ferroelectric material has been well known in the field even for the bulk crystal from early time – Barkhausen pulses (For example, Barkhausen Pulses in Barium Titanate, Chynoweth. Phys. Rev. 110, 1316 (1958)).

Response to comments:

We carefully considered the reviewer #1's comments in the last review and revised the manuscript according to the comments. The reviewer has newly commented that our paper is outdated in this review. Yes, several papers have been published including the review type of papers very recently which we did not include in the reference. This might be partly due to the fact that we have been taking a time for the revision. This is our fault, but it is true that some of them had not been published when we submitted it. Nevertheless, we should have updated understanding of negative capacitance (NC), referring necessary papers published recently. Thanks for suggesting these papers. Even with updating, however, we still think our work definitely provides consistent data and discussion which distinct from those published very recently in literatures on NC.

As the reviewer pointed out, transient and quasi-static “NC” effects have been intensively discussed. However, each model is based on traditional understanding of ferroelectric materials, and it is different from other ones in terms of polarization switching kinetics. This means no consensus on NC has been reached yet. Particularly, when the case of multi-domain FE is concerned, the issues whether quasi-static NC effect can be stabilized or not and how it can be stabilized are still under intensive debate. In fact, a critical comment has been pointed out in J. Iniguez's review paper as “*Thus, further device modelling that explicitly considers multidomain gates with mobile domain walls is required.*” Although both W. H. Park and I. Luk'yanchuk proposed the model of domain wall motion to predict quasi-static NC in multi-domain system, the domain shape and resultant NC effect are different from each other. Moreover, both are conceptual models without any confirmation of experiments results.

Meanwhile, our paper has a novelty and significance as follows.

- 1) We have proposed a model based on the experimental data taken with the careful experimental setup for the first time.
- 2) Although the charge dynamics in our model is partly similar to those in W. H. Park and I. Luk'yanchuk's papers, our results and model show a critical difference from the quasi-static NC effect discussed in these papers in terms of the fact that continuous change of polarization is

impossible. It is the key to discuss the confused debate on quasi-static NC effect as we discussed in section IV.

3) As the reviewer#1 commented, the zig-zag behavior of voltage response is not expected by those models. Although a recent publication after our paper submission reported the zig-zag behavior experimentally, a relation between zig-zag jumps and polarization switching was not made clear. It is because quantitative results cannot be obtained without careful experimental setup. Our results, however, for the first time, clearly have shown the zig-zag behavior at $V_F \sim \pm V_C$ based on carefully designed experiments, providing a reliable model for NC effect.

4) Although Barkhausen pulses have been observed in bulk ferroelectric crystals in early days, they are derived from general domain motion and the zig-zag behavior observed near V_C in the present FE/DE stack is definitely different.

In short, despite many papers about NC effect have been published recently, our paper, for the first time, has experimentally demonstrated the successive stepwise internal voltage jumps along with multiple domain switching in a FE/metal/DE system under well designed experimental setup. Based on the results, we have quantitatively modeled the internal voltage boosting. This is different from NC models proposed so far conceptually and is critical for understanding the steep SS in so-called NCFET. We believe these results significantly contribute to both materials science and electron device communities.

Since it is true that the reviewer's comments have actually stimulated us to further consideration of the polarization switching kinetics, we would like to thank the reviewer and we would make our above novelty more clear in the text in addition to add appropriate new citations (Ref. 9-11, 31, 32).

Introduction:

...

Several models of quasi-static NC associated with domain wall motion in a multiple-domain system have been also proposed^{9, 10-11}.

...

However, physical understanding of NC effects is still under intensive debate¹⁹⁻³¹. Experimentally observed NC effects are different from each other, and also from the concepts initially proposed. Alternative explanations for the capacitance enhancement and transient NC effects in FE/PE stacks have been also proposed²⁰⁻²⁶. For example, a feasibility of capacitance enhancement is explainable from a strong coupling between FE and PE layers²⁰⁻²², while the transient NC is understandable from the viewpoints of overshoot in voltage supply or slower speed of charge compensation at FE/PE interface relative to polarization switching in FE film²⁴⁻²⁶. In fact, it has been argued that NC region of FE material is intrinsically unstable or even impossible^{19, 27}. In addition, the SS improvement observed in a FET mostly suffers from critical problems that a large hysteresis is detected, a high voltage is needed and an operation frequency is limited in actual experiments²⁸⁻³⁰.

To sum up, experimental evidences provided so far are insufficient to conclude the concept of quasi-static NC, and reliable modelling of SS improvement in a FET with FE/PE stacks are still missing. These should be verified urgently, because they are critical for further advancing the material science as well as electron device performance of FE/PE stacks to elucidate whether the quasi-static NC can be really stabilized or not, and whether the steep SS characteristic so far demonstrated is really promising for low-power CMOS applications or not.

Section III:

Note that the resulted zigzag $Q-V_F$ characteristics is totally different from the conventional $Q-V_F$ curve in a single FE-CAP or from the S-curve expected from the NC theory¹. It is also significantly different from the characteristics expected from the recent models with the multiple-domain system,

in which a continuous change of polarization with the help of domain wall motion is assumed⁹⁻¹¹.

Overall, the observed internal voltage boosting could be understood in a very straightforward manner. When a sufficiently high voltage (to make $V_F > V_C$) is applied, a certain domain flips, and the corresponding amount of charge varies. This amount of charge is too large to be compensated by the PE layer, so the only method not to violate the charge conservation law is to increase the voltage across the PE layer over the applied voltage. This is especially the case near V_C . Then, the voltage across the FE layer must be dropped. This is all that the authors have observed from their test setup. Besides, similar experiments have been performed by other authors (Jo et al., Negative Capacitance in Organic/Ferroelectric Capacitor to Implement Steep Switching MOS Devices, Nano Lett. 2015, 15, 4553–4556) although the authors of that work have completely misinterpreted their results. The authors of the present manuscript used a better experimental setup and did not fall into the wrong conclusion. Nonetheless, the experimental observations are fundamentally identical to the other work.

Response to comments:

Yes, our experiments look similar to those reported by Jo et al. in Nano Lett. However, as we mentioned above, the internal voltage has not been analyzed by careful measurements, and so the internal potential jumps corresponding to $\pm V_C$ of FE film has not been characterized in Jo et al.'s paper. In fact, Fig. 3(a) in Jo et al.'s paper (shown below) indicates that V_F corresponding to V_{int} jump (obtained by subtracting V_{int} from V_G) was -1.2V (forward) and -2V (backward) for the green line, while V_C is 2.3 V and -1 V as pointed out in the paper.

Without showing internal potential jumps at around $V_F \sim \pm V_C$, we cannot conclude that the jumps are associated with domain switching straightforwardly as you understood above. This is critically an important point to clarify domain switching kinetics in conjunction with internal voltage boosting. And it is different from models of quasi-static NC reported very recently. So, we do not think our experimental observations are identical to other works. We would be very happy if the reviewer could understand our standpoints and this work value.

Nevertheless, thanks for suggesting Jo's paper which we missed before. We have cited it (Ref. 32) and stated our standpoint on this issue more clearly in the introduction part as:

A direct way to examine the actual NC effect in FE/PE stack is to investigate the voltage at the internal node, V_{int} , between FE and PE layers in DC mode, which makes possible of the direct comparison of V_{int} with SS. In fact, a couple of works on internal potential measurement have been

reported, but they have only qualitatively discussed about this issue, resulting that a consistent model could not be provided^{12,32}. We have suspected it might be due to experimental difficulties of measuring the internal potential in FE/PE stack. Therefore, in this work, accurate DC measurements are particularly paid careful attention to. The stepwise V_{int} jumps at the coercive voltage, $\pm V_C$, of FE layer in FE/PE stack are demonstrated, and a relationship between V_{int} jumps and the steep SS in FET with FE/PE gate stack is presented. They are quantitatively understood from the viewpoints of successive polarized domain flipping and depolarization feedbacks from the PE-CAP. The results provide a clear physical insight to understanding the small SS values in FETs with FE/PE gate stacks reported so far.

Another critical conceptual problem of this work is as follows: the authors compared the MFM-MIM (or MIS, e.g., MFMIM or MFMIS) with MFIM (or MFIS) systems. These two systems are completely different regarding the boundary conditions. In the former case, the intervening metal layer provides a sufficient compensating charge in response to the change in the ferroelectric bound charge. Also, any (local) change in the FE layer, such as local domain reversal, is responded by the entire PE layer due to the presence of the intervening metal layer, which shorts out the surfaces of the FE and PE layers. In contrast, the latter case has a completely different behavior; the FE and PE layers must respond to the external stimuli together as a single body since there is no intervening metal layer that separates the two. Please refer to Hoffmann et al.[Nanoscale 2018, 10, 10891]. Therefore, the direct comparison made in Fig. 4 does not make any sense.

Response to comments:

The reviewer has pointed out the difference of compensating charges between MFMIM and MFIM structures. First of all, the initial concept of “NC” effect is derived from the Landau theory with a single domain FE film. It has also been discussed in Hoffmann’s paper that the MFIM system is identical to MFMIM one if FE layer is with a single domain. In fact, we think the interface charges can follow the polarization switching even in MFIM structures in the quasi-static or not-so-fast switching mode, because electron-hole pairs should also be formed to accommodate the Gauss's law at F/I interface in the case of single domain. Therefore, we compared our model in the case of single domain with original concept of NC. We think it is quite fair. We have made this point more clearly in the paper as below:

The intrinsic difference between our view and the original proposal on NC effects in FE-based FETs is next discussed. To be fair, only the case of single domain is considered.

Meanwhile, we agree that in multi-domain structure case, the charge compensation (screening) process should be obviously different between two stacks in terms of non-uniform charge compensation. On the other hand, in the actual MFIS case, the charge compensation between neighboring domains at the interface should also be considered, which makes the situation more complicated even if the phonon coupling between two layers is not taken into consideration. In this revision, we have added a comment on this point as below:

Finally, it is worthy of mentioning the following two points for the multiple-domain system. First, the polarization kinetics depends on a specific model such as Kolmogorov-Avrami-Ishibashi or nucleation-limited switching models^{26,34}. But this is beyond this paper in which the DC-mode polarization switching rather than high-speed switching has been focused. When the polarization switching speed is concerned in device applications, the specific domain switching kinetics should be taken into account for NC effect analysis. Second, there will be a difference between the case w/ and w/o internal electrode. In the latter case, the charge flow at FE/PE interface, the local effect of domain switching and the coupling effects between FE and PE layer should be taken into

consideration^{11, 20-22}. It is very interesting to further study these issues.

Moreover, the result from the load line analysis of MFMIM structure in Fig. S4(b) would not be directly applied to the NCFET operation. Since a semiconductor has a highly nonlinear capacitance depending on the voltage region, the hysteresis engineering by modulating the external capacitance would be highly complicated than this simple scheme presented in Fig. S4.

Response to comments:

Concerning Fig.S4(b), yes, hysteresis engineering in real transistors is much more complicated due to the variable capacitance effect in semiconductor. We have commented this point in our initial text as “*Meanwhile, it is easy to expect that this will be practically very challenging when the bias dependence of semiconductor capacitance in FET is considered*”. And we have also modified the related content about this point in supplementary material, making it more clearly as below:

....

To find out how to control the subthreshold hysteresis in FET with FE/PE gate stack, the biggest V_{int} gain is focused. Since C_P and P_1 dominantly affect the position and the magnitude of V_{int} gain, the hysteresis in FET might be minimized by achieving an overlapping region of V_{int} gain in both forward and backward sweepings if the bias dependence of semiconductor is ignored, as schematically shown in Fig. 4S(c)

...

As a matter of fact, it is not easy but rather impossible to control the polarization and switching of each domain. In addition, the coercive field in each domain is not identical and the semiconductor capacitance depends on the electric field. Thus, hysteresis-free V_{int} gain and Q- V_F characteristic are obviously difficult to be achieved in actual FE/PE gate stack FETs.

In this paper, we have mainly focused on the bound charge kinetics in FE/DE system through the internal potential measurements. And we have intended to show a technical direction about how to reduce the hysteresis. For actual FET applications, we would optimistically expect that the bias dependence of capacitance in semiconductor may be smaller in fully depleted channel such as FinFETs and ultra-thin SOI FETs.

While the authors’ experimental setup to exclude the charge exchange between the internal node and ground through the V-meter, which make it possible to estimate the variation in the node voltage in a quasi-static manner, is plausible, the presented arguments do not contribute that much to the current understanding of the field.

Response to comments:

It’s a pity if we may not provide the reviewer#1 with new insight into NC. As we mentioned above, despite a number of papers have been published, complete understanding of NC effects is still under debate. We know that apparently similar experiments and models have been reported in recent publications, while there are also try-and-error types of experiments and conceptual models to describe NC effect and to demonstrate sub-60mV/dec in subthreshold swing. This is why we have carefully studied this issue experimentally to provide the community with reliable and clear messages. Careful experiments are essential for obtaining meaningful results, which can only clarify physics behind. Our paper has consistently demonstrated V_{int} jumps at $V_F = \sim \pm V_C$ in a stepwise manner for the first time, based on careful experimental setup. On the basis of this, we present a reliable model which is different from those proposed conceptually. We believe our consistent results and models provide clear and reliable information on a state of confusion regarding NC effects and NC-FETs.

Reviewer #2 (Remarks to the Author):

The authors have very carefully addressed all the comments and remarks. They have also improved the text of the article and the references.

Especially, they explained how they considered the effect of impedance in the measurement system for accurate measurement of internal potential in high impedance system. In the new version of revised manuscript, they usefully added further consideration about the effect of leakage through FE-CAP. All these clarifications are very useful and improve the trust in the result.

Response to comments:

Thanks for positive comments on our last revision and for your careful reviews in the 1st and 2nd runs.

Overall, the paper offers a very good insight into the zig-zag modeling in polarization behavior in polycrystalline PZT capacitors and there is indeed a difference between the author's view taking into account domain flip and the original proposal of NC effects in FE-based FETs by Salahuddin. This is still debatable and to be investigated in different ferroelectric material systems and compare the various models.

Response to comments:

Thanks for helpful comments. We agree that the quasi-static NC effect is still controversial though new models of domain wall motion have been proposed. Our results and model are different from both Salahuddin's concept and recent models, while we do not mean the quasi-static NC is impossible. More careful investigation is needed. In addition, we also agree that specific behavior of so-called NC effect may depend on the ferroelectric material systems. Other material system should be also investigated and compared. This is our research plan in next step.

In conclusion, while I still strongly believe this paper is very focused technically and does not have a high level of novelty (the zig-zag effect itself is not new and the novelty part is about modelling and discussion of this effect) it includes very useful data and discussions to advance the field of negative capacitance. Therefore, I agree with the publication, provided that the Editors considers this novelty on modelling enough.

Response to comments:

We would appreciate your agreement of the publication. Concerning the novelty, although zig-zag effect has been reported in a paper, our results demonstrated, for the first time, that the zig-zag internal potential jumps occurs at $V_F = \sim \pm V_C$. This is thanks to careful set-up of the measurement system. And, it is critical to get a solid physical understanding. We would be very happy if you could understand our new points in experiments and results in addition to the modeling.

Reviewer #3 (Remarks to the Author):

The reviewer is satisfied with the authors' response to comments 1, 2, 4 and 5. However, the reviewer has several further comments/questions concerning #3, as follows:

1. In Fig. S3, the curve corresponding to $C_p=1.5\text{nF}$ is identical to the curve in Fig. 2(d), which, as described in the text, is the result when $C_p=0.5\text{nF}$. This obvious inconsistency may be a small mistake, but the reviewer still hopes that the authors could correct this and provides an explanation.

3. The experimental results shown in Fig. S3 are mysterious. The reviewer compared the curves for $C_p=0.1\text{nF}$ and $C_p=15\text{nF}$, and made these observations

(1). Considering the $C_p=0.1\text{nF}$ case: In the forward sweeping non-switching region, V changes from -5V to -2V , and the V_F changes from -1V to 1.5V , which means almost all the applied voltage drops across the FE. In this case, the RC time constant is $\sim 5\text{s}$

(2). Considering the $C_p=15\text{nF}$ case: In the forward sweeping non-switching region, V changes from -6V to 4V , and the V_F changes from -1V to 1.3V . This time, most of the applied voltage drops across the DE. The RC constant is $\sim 750\text{s}$.

In the non-switching region, based on the capacitance-voltage divider rule, a larger C_p gives rise to a higher V_F . Also, a larger RC time constant means that the measurement is less affected by the leakage, such that the results should be closer to the ideal capacitance- voltage divider case. But the results shown in the figure are clearly not consistent with the above analysis.

Response to comments:

We would thank reviewer #3 very much for pointing out our careless mistake in Fig. S3. We marked 0.1nF to 15nF in the reversed order in Fig. S3. We would like to sincerely apologize for the low-level mistake. We have corrected the figure as shown below. The red curve should correspond to $C_p=0.5$ and consistent with the curve in Fig. 2(d).

Comments #3 also relates to the mistake in Fig. S3. In the correct figure as shown above, the dark green curve is for the case of $C_p=15\text{ nF}$. It makes sense that most of the applied voltage drops across the FE layer in case of $C_p \gg C_F$. Namely, when V changes from -5V to -2.2V , V_F changes from -1V to 1.6V . Meanwhile, the dark-red curve is for the case of $C_p=0.1\text{nF}$. It also makes sense that most of the applied voltage drops across the PE layer in case of $C_p < C_F$. When V changes from -6V

to 4V, the V_F changes from -1V to 1.3V

We are very sorry again for the wrong indication of C_P on the curves in the figure.

2. In their response, the authors admitted that the leakage of the FE capacitor was not negligible, and thus in the non-switching region the measured V_{int} and V do not follow the CFE-CDE voltage divider rule. They calculated that the RC constant was roughly 5s. If this is indeed the case, then the reviewer has the following critical concerns.

- (1). The author said that the data was taken after ~ 1 s. Compared to the RC time constant of 5s, ~ 1 s is a relatively short time during the transient when the rate of change is large, and any variation of the time used to take the data could significantly affect the measured V_{int} .
- (2). With this leakage path, the internal node is no longer isolated, and thus the net charge on the internal node can accumulate during the measurement.
- (3) During polarization switching, both the capacitance and the resistance of the ferroelectric capacitor become nonlinear, and thus the RC time “constant” during polarization switching is not a constant. Also, due to the randomness of the polarization switching according to the nucleation-limit-switching model, the resistance and capacitance may have some variations during switching. Based on these concerns, it is reasonable for the reviewer to suspect that the experimental results are not sufficiently solid to support the conclusion of this article. The zigzag curve during polarization switching could be caused by the measurement itself, or the randomness of the RC time constant during switching, instead of the explanation based on the Landau theory.

Response to comments:

We would thank the reviewer for the careful remarks on RC time constant in the sample system. We respond to the above sub-comments one by one.

- (1) The reviewer’s concern may be that it is not easy to pick up the voltage changing very rapidly at a relatively short time (1s). In our experiments, it was actually not difficult to read it, probably due to the fact that the time constant is not a simple time constant in an exponential type as suggested in comment (3).
- (2) Yes, the net charge on the internal node is changing with time due to a leakage path in FE film, so the internal voltage is also changing in our measurement. Yes, but it is not relevant in our measurements, because we are not interested in V_{int} but in $V-V_{int}$. Both measured values of V and V_{int} are correct within our experimental accuracy as mentioned in our paper, but physically no meaning. While, $V-V_{int}(=V_F)$ is physically relevant and more important for discussing the domain switching in the FE film.
- (3) The time constant of the system depends on the applied voltage due to a non-linear conductance and capacitance of FE-CAP as pointed out. However, it is not true that “*the zigzag curve during polarization switching could be caused by the measurement itself, or the randomness of the RC time constant during switching*”. Our results have shown that the zigzag voltage jumps after the first one occur very near to $V_F=\pm V_C$. A small deviation from V_C observed experimentally might come from measurement errors as pointed out by the reviewer, or due to domain-domain interaction as we commented in the paper. But the randomness of the time constant cannot provide the stepwise voltage jump but should lead to a more continuous influence on the V_F - V curve. In addition, since even a small V_F change should actually cause a huge charge movement near V_C in the FE film, the zigzag behavior clearly observed in the experiments neither can be explained by experimental system errors nor by unknown trapped charge effects. Thus, only a possible other way to explain the results may be by “an accidental randomness”. In that case, we will not be able to see these results twice. As we have discussed in the paper, this is not the case in the present experiments. Thus, we are quite sure that experimental results are sufficiently solid to provide the conclusion.

We hope you will be satisfied with the present revision. And, we would appreciate your helpful comments in the first and second runs of reviews.

Sincerely

Xiuyan Li

Shanghai Jiao Tong University

3rd Review on the manuscript “Stepwise internal potential jumps caused by multiple-domain polarization flips in metal/ferroelectric/metal/paraelectric/metal stack” by Xiuyan Li, and Akira Toriumi.

In this second revised manuscript, the authors mainly revised their explanation more clearly. We believe that most of the inconsistencies or limitations of their models are well addressed and clearly noticed in the manuscript. While we still agree with reviewer #2’s standpoint that the novelty of this study is not satisfactory for the publication, we value the authors’ insight on the zig-zag modeling of polarization and the consistent experimental results by the careful measurement. Therefore, we agree with the publication when the following issues could be appropriately addressed.

1. In response to the first comment, the authors revised the introduction part by addressing the recent multi-domain negative capacitance models. To our knowledge, among the numerous papers on the negative capacitance, only a few of them have dealt with the multi-domain model so far. (e.g., I. Luk’yanchuk et al. [Phys. Rev. B 2018, 98, 024107], [Commun. Phys. 2019, 2, 1.]), and comprehensive review papers (J. Iniguez et al., Ferroelectric Negative Capacitance, *Nature Reviews*, 2019, and H. W. Park et al., Modeling of Negative Capacitance in Ferroelectric Thin Film, *Advanced Materials*, 2019). We believe all of them are very crucial to improve our understanding of the NC effect. However, in this revised manuscript, the last article (namely, H. W. Park et al.) was missing, and also, the authors wrongly mentioned the name of the author as “W. H. Park” in the response letter.

2. The authors stated that their model does not necessarily require a specific domain switching kinetics of the ferroelectric thin film as follows,

“Although the physics behind with regard to the bound charge movement is similar to the present work, our model does not necessarily require a specific domain switching kinetics in terms of whether it is like KAI, NLS or statistical-KAI or -NLS models, because the DC-mode polarization switching rather than high-speed switching has been focused in this paper as mentioned above.”

However, associated with reviewer #3’s previous comment,

Reviewer #3 :

“This paper advances the understanding of the so-called “negative capacitance” phenomena that have attracted a great deal of interest in the last few years. The authors treat the multi-domain ferroelectric thin film under discussion as a combination of small Landau domains, with each domain following the Landau theory of ferroelectricity.”

this statement naturally arises the following question: Does this model assumes the continuous switching (i.e., homogeneous switching across the Landau barrier) for each domain, (and the multi-domains of this continuous switching with a distribution

of the coercive fields) or it excludes this mechanism and sticks to the classical switching kinetic by the nucleation and growth (without involvement of the “Landau barrier”). We believe that the fundamental assumption on the switching kinetics of each domain is a quite critical factor to determine the desired hysteresis free operation. (See for example, Kim et al., Nano Lett. 2017, 17, 7796–7802, “Voltage Drop in a Ferroelectric Single Layer Capacitor by Retarded Domain Nucleation”) We recommend the authors to clarify their standpoint on this issue, and discuss a possible difference between the resulted stepwise voltage drop from each switching mechanism.

3. We agree with reviewer #3 that this particular experimental set-up for measuring the internal voltage without the leakage through the voltmeter would be very helpful to an experimentalist who attempts similar measurements. However, the answer to the reviewer #3’s comment, namely, the detailed explanation of the measurement set-up still seems insufficient. We highly recommend the authors to specify more clearly their experimental devices (e.g., the high impedance voltmeter with a high precision current preamplifier the authors used for the experiment), and implementation set-up with the conventional DC measurement device.

Reviewer #2 (Remarks to the Author):

The authors made a significant effort to address and discuss every point raised by the reviewers. In the current form, and taking into account the answers of the authors, the paper has the merit to propose a credible and experimentally supported modeling of the zig-zag phenomena in multi-domain ferroelectrics in relation with the concept of negative capacitance.

I support the publication of the paper with some final minor revisions that should be checked only by the Editor, according to the remarks below.

1) Make clear in the abstract that the novelty is about the modeling and the validation of the model by experimental data. I am less sure that the claim made that this paper 'also provide new insights into SS engineering in FE-based FETs. The authors have not provided such path. They are just investigating, modeling and explaining bound charge emission associated with each domain flip in multiple-domain FE/PE stacks.

2) Cite more properly the published literature on NC devices using single and multi-domain PZT where similar effect have been reported. Without insisting with such list, there are significant number of relevant paper in recent IEEE IEDM and IEEE journals (EDL and TED) with similar experimental results. We encourage the authors to make a full and fair citation of existing literature, which is not yet the case. In some papers even near-hysteresis free devices have been reported and also the correlation between the internal gain and the steepness. The authors should comment the consistency of their results with the results of such comprehensive literature on NC in PZT.

3) The authors clarified the role of the leakage and the removed any doubts on the electrical setup. They are discussing and characterizing the characterizing internal charge kinetics and the reviewer agrees with them that specific domain switching kinetics should be taken into account for NC effect analysis. Two remaining suggestions:

(1) report the leakage current level in normalized way (per cm²) in your capacitors, for instance Fig S2 where the leakage characteristics of single PZT-CAP are reported, and compare to state of the art PZT leakage.

(2) What is missing from the paper in this final form is a multiple time repeated experiment (3 to 5 times) and carefully looking at the zig-zag evolution in same measurement conditions (when repeated) and a discussion of the observed repeatability or changes based on the model, as the authors are referring to randomness of the switching process.

Reviewer #3 (Remarks to the Author):

The reviewer is satisfied with the responses to comments #1 and #3, but there remains some concerns about the response to comment #2.

Although the reviewer agrees that even with this leakage, the measured V_{int} and calculated V_{FE} could both be correct as the author indicates, it is still not clear how this leakage current would affect the observation of the apparent "negative capacitance" effect; i.e., in order to be consistent with the "negative capacitance" effect, one should observe that $[V]_{int}$ rises faster than V_{APP} , but the author has failed to demonstrate whether the leakage current has erroneously caused this "negative capacitance" feature.

Reviewer #1

3rd Review on the manuscript “Stepwise internal potential jumps caused by multiple-domain polarization flips in metal/ferroelectric/metal/paraelectric/metal stack” by Xiuyan Li, and Akira Toriumi.

In this second revised manuscript, the authors mainly revised their explanation more clearly. We believe that most of the inconsistencies or limitations of their models are well addressed and clearly noticed in the manuscript. While we still agree with reviewer #2’s standpoint that the novelty of this study is not satisfactory for the publication, we value the authors’ insight on the zig-zag modeling of polarization and the consistent experimental results by the careful measurement. Therefore, we agree with the publication when the following issues could be appropriately addressed.

Response to comments:

First of all, thanks for positive comments on the last revision and for your careful reviews in the 1st, 2nd and 3rd runs.

1. In response to the first comment, the authors revised the introduction part by addressing the recent multi-domain negative capacitance models. To our knowledge, among the numerous papers on the negative capacitance, only a few of them have dealt with the multi-domain model so far. (e.g., I. Luk’yanchuk et al. [Phys. Rev. B 2018, 98, 024107], [Commun. Phys. 2019, 2, 1.]), and comprehensive review papers (J. Iniguez et al., Ferroelectric Negative Capacitance, Nature Reviews, 2019, and H. W. Park et al., Modeling of Negative Capacitance in Ferroelectric Thin Film, Advanced Materials, 2019). We believe all of them are very crucial to improve our understanding of the NC effect. However, in this revised manuscript, the last article (namely, H. W. Park et al.) was missing, and also, the authors wrongly mentioned the name of the author as “W. H. Park” in the response letter.

Response to comments:

We would like to apologize very much for making a spelling mistake on the author’s name “H. W. Park” in the last response letter.

Concerning the literatures on the multi-domain model, yes, as you said all of these (also Hoffmann’s paper modelled multiple-domain case) are very crucial to improve our understanding. We had read H. W. Park’s paper very carefully. It shows an interesting modeling of NC effect in multi-domain system in addition to a comprehensive review of NC studies. We did not cite it in the previous revision, just because it was published after our paper submission. We mentioned a similar fact to the editor in the first revision to be fair. Nevertheless, according to your and reviewer #2’s suggestions, we have included this reference (Ref. 12) as well as other two articles published after our paper submission (Ref. 16, 24) in the present revision.

2. The authors stated that their model does not necessarily require a specific domain switching kinetics of the ferroelectric thin film as follows,

“Although the physics behind with regard to the bound charge movement is similar to the present work, our model does not necessarily require a specific domain switching kinetics in terms of whether it is like KAI, NLS or statistical-KAI or -NLS models, because the DC-mode polarization switching rather than high-speed switching has been focused in this paper as mentioned above.”

However, associated with reviewer #3’s previous comment,

Reviewer #3: *“This paper advances the understanding of the so-called “negative capacitance” phenomena that have attracted a great deal of interest in the last few years. The authors treat the multi-domain ferroelectric thin film under discussion as a combination of small Landau domains, with each domain following the Landau theory of ferroelectricity.”*

this statement naturally arises the following question: Does this model assumes the continuous switching (i.e., homogeneous switching across the Landau barrier) for each domain, (and the multi-domains of this continuous switching with a distribution of the coercive fields) or it excludes this mechanism and sticks to the classical switching kinetic by the nucleation and growth (without involvement of the “Landau barrier”). We believe that the fundamental assumption on the switching kinetics of each domain is a quite critical factor to determine the desired hysteresis free operation. (See for example, Kim et al., Nano Lett. 2017, 17, 7796–7802, “Voltage Drop in a Ferroelectric Single Layer Capacitor by Retarded Domain Nucleation”) We recommend the authors to clarify their standpoint on this issue, and to discuss a possible difference between the resulted stepwise voltage drop from each switching mechanism.

Response to comments:

Thanks for an invaluable comment.

In section III of our paper, we have modelled our results electrostatically instead of using Landau or other kinetic models by assuming that the intermediate state between two polarized states is not stable even quasi-statically. In the case of multiple-domain FE/PE system, we have assumed that each domain switches independently and successively, causing a zig-zag $Q-V_F$ characteristics as a whole. For each domain, our model does not necessarily require a specific switching kinetics (such as homogeneous switching or nucleation and growth), because when the movement of bound charges associated with the domain switching forms the depolarization field on FE layer, the internal voltage gain can be obtained in any switching kinetics. In fact, this view has also been pointed out in H. W. Park’s review paper (*Nonetheless, such domain mechanism does not necessarily correspond to the total frustration of the internal (differential) voltage boosting effect.*). Meanwhile, we agree that such properties as the gain factor and/or time-dependence of V_{int} , could be affected by a specific switching mechanism. But this is beyond our discussion in the present work, because our experiments were carried out in DC mode. We are actually interested in investigating kinetic/dynamic effects it in the next step.

In section IV, we discussed the difference of our model (in the case of single domain) from the original idea of NC which is based on the assumption of homogeneous Landau switching with a single domain. The G-P diagram with bi-stable states corresponding to $\pm P$ should be similar for two cases except the barrier height (Kim’s paper also described energetics in such a way). So, we also used the double-well energy diagram to compare with the original NC theory (Fig. 5). This might mislead the reviewer #1 into the fact that we treated each domain as homogeneous Landau switching. We are sorry for not explaining our viewpoint clearly in the response of the first revision.

We would appreciate the reviewer #1 for very helpful suggestions and comments on this unclear description. In this revision, we have slightly modified Fig. 5 for avoiding readers' misunderstanding and revised section IV by stating above viewpoint more clearly as below.

Finally, it is worthy of mentioning that the polarization switching kinetics depends on a specific model such as Kolmogorov-Avrami-Ishibashi or nucleation-limited switching models^{31,41}, while specific switching kinetics does not necessarily lead to the total frustration of the V_{int} enhancement effect¹². Namely, when the depolarization field is formed due to the bound charge movement, the V_{int} gain can be obtained in any switching kinetics cases. Meanwhile, the gain value, shape and time dependence of V_{int} , might be affected. But such consideration is beyond the scope of this paper in which the DC-mode polarization switching rather than high-speed switching is focused. When the polarization switching speed is concerned in device applications, the specific domain switching kinetics should be taken into account for the NC effect analysis.

Fig. 5 The difference between our view and the original NC proposal. (a) In our view, V_{int} gain is associated with the polarization flip, and the intermediate state between two polarization states is assumed to be unstable. Thus, $V_F > V_C$ is required to get V_{int} gain. (b) In the original NC proposal, the intermediate state can be stabilized by changing the polarization value. Then even $V_F=0$ enables a V_{int} gain directly.

3. We agree with reviewer #3 that this particular experimental set-up for measuring the internal voltage without the leakage through the voltmeter would be very helpful to an experimentalist who attempts similar measurements. However, the answer to the reviewer #3's comment, namely, the detailed explanation of the measurement set-up still seems insufficient. We highly

recommend the authors to specify more clearly their experimental devices (e.g., the high impedance voltmeter with a high precision current preamplifier the authors used for the experiment), and implementation set-up with the conventional DC measurement device.

Response to comments:

Thanks for a kind suggestion. We used Keithley 6430 source-meter unit equipped with a remote pre-amplifier to measure the internal potential. This preamplifier enables a current flow to the measurement system to be down to sub-fA level. Thus, the input impedance increases up to $10^{16} \Omega$ effectively. We have added the details in the method part as below.

A DC-voltage measurement system (Keithley 6430) equipped with a remote sub-fA pre-amplifier was connected to the internal metal node between FE and PE CAPs to measure the internal potential. This enabled us to effectively increase the input impedance of the measurement system up to $10^{16} \Omega$.

Reviewer #2 (Remarks to the Author):

The authors made a significant effort to address and discuss every point raised by the reviewers. In the current form, and taking into account the answers of the authors, the paper has the merit to propose a credible and experimentally supported modeling of the zig-zag phenomena in multi-domain ferroelectrics in relation with the concept of negative capacitance. I support the publication of the paper with some final minor revisions that should be checked only by the Editor, according to the remarks below.

Response to comments:

Thanks for positive comments on the last revision and for your careful reviews in the 1st, 2nd and 3rd runs.

1) Make clear in the abstract that the novelty is about the modeling and the validation of the model by experimental data. I am less sure that the claim made that this paper 'also provide new insights into SS engineering in FE-based FETs. The authors have not provided such path. They are just investigating, modeling and explaining bound charge emission associated with each domain flip in multiple-domain FE/PE stacks.

Response to comments:

Thanks for invaluable suggestions. We have revised the abstract to provide a clear message on the modeling and the validation of the model as below.

In this work, stepwise internal potential (V_{int}) jumps observed near the coercive voltage of FE layer in FE/metal/PE system are reported in carefully designed DC measurement system. A relationship of this effect with the steep SS in FET is also confirmed experimentally by connecting a FE capacitor to a simple metal-oxide-semiconductor FET. On the basis of the experimental results, it is analytically modelled that the observed V_{int} characteristics are derived from the bound charge emission associated with each domain flip in multiple-domain FE/PE stack. This view is different from the original concept of NC and will provide a useful guidance to practically characterize FETs with FE/PE gate stacks.

In addition, we mentioned “*also provide new insights into SS engineering in FE-based FETs*” from the following viewpoint. Our model is different from the original view of NC proposed by Salahuddin and Datta. However, the steep SS could be achieved and the hysteresis could be engineered in principle even with our model, though it was a bit difficult practically to achieve the perfectly hysteresis-free steep SS FET. We have discussed this point in Section III as shown in the response to your comment 2). Therefore, we think our paper provides new insights into the SS engineering in FE-based FET. We would appreciate your comments and for providing us with a chance to modify the statement to be clearer.

2) Cite more properly the published literature on NC devices using single and multi-domain PZT where similar effect have been reported. Without insisting with such list, there are significant number of relevant paper in recent IEEE IEDM and IEEE journals (EDL and TED) with similar experimental results. We encourage the authors to make a full and fair citation of existing literature, which is not yet the case. In some papers even near-hysteresis free devices have been reported and also the correlation between the internal gain and the steepness. The authors should

comment the consistency of their results with the results of such comprehensive literature on NC in PZT.

Response to comments:

Thanks for suggestions. We have cited some papers which demonstrated similar experiments to ours, namely the internal potential measurement in DC mode, such as Jo, J. et al. Nano Lett. 15, 4553-4556 (2015). In the first revision, we did not cite Saeidi, A. et al.'s paper published in Scientific Report, because it was published after our paper submission. According to suggestions from you and the reviewer #1, we have added this paper in the present revision. In addition, we are sorry for still missing some important papers about NC effect of PZT in recent IEEE conferences and journals. Now we have found important papers, including a very recent one (Verhulst, A. S. et al. TED 63, 377-382 (2020), Saeidi, A. et al. EDL 38, 1485-1488 (2017) and Saeidi, A. et al. IEDM (2018)). We have discussed the difference and consistency between our results and those in these literatures at appropriate position as below.

Section I

Note that the V_F drop along with the V_{int} jump occurs very near $\pm V_C$ in FE-CAP. This fact is critically important from the viewpoint that it is beyond qualitative observation in the similar measurements reported recently^{16,37}, because it directly indicates that the V_{int} gain is associated with the polarization flip in a FE layer.

Section III

Hysteresis of the V_{int} gain is critical for achieving steep SS FETs in advanced CMOS. According to our modelling, nearly hysteresis-free V_{int} gain is made possible in principle by inserting an appropriate C_P in the case of the single domain FE in FE/PE stack. The requirement for this condition is the same as that for stabilizing NC effect originally proposed. This is partly consistent with a recent report in which nearly single domain PZT film together with strict capacitance matching was employed and a nearly hysteresis-free steep SS was demonstrated^{23,24}. In the case of FE layer containing multiple domains, the hysteresis-free V_{int} gain is difficult to achieve perfectly. Here, let us focus on the biggest V_{int} gain. Since the V value and amplitude of V_{int} jump are controlled by C_P and P_1 , respectively, it becomes possible that the V_{int} jump in the forward V sweeping may overlap with that in the backward one by adjusting P_1 under an optimum C_P , (**Fig. 4(c)**). Resultantly, the subthreshold hysteresis in FETs should be substantially reduced in the overlapped range. As a matter of fact, the bias dependence of semiconductor capacitance makes hysteresis-free FET more difficult. Thus, hysteresis is seen in most of the steep-SS FETs reported up to now^{16,22,23}.

3) The authors clarified the role of the leakage and the removed any doubts on the electrical setup. They are discussing and characterizing the characterizing internal charge kinetics and the reviewer agrees with them that specific domain switching kinetics should be taken into account for NC effect analysis. Two remaining suggestions:

(1) report the leakage current level in normalized way (per cm²) in your capacitors, for instance Fig S2 where the leakage characteristics of single PZT-CAP are reported, and compare to state of the art PZT leakage.

Response to comments:

Thanks for nice suggestions. Normalized leakage current is shown in Fig. S2(c) in this revision. We purchased PZT samples from Radiant Technologies Inc. and the film thickness was ~500nm. The leakage current density was below 10^{-8} A/cm² under 1V (20KV/cm). This is comparable to the level reported for the state of the art PZT (Ref 39, 40). We have added a comment in the text as

The capacitance at the center of C - V_F characteristics was ~0.25 nF and the leakage current density at 1 V was below $\sim 10^{-8}$ A/cm², which is comparable to the level reported for the state of the art PZT^{39,40} and corresponds to 5×10^{10} Ω .

(2) What is missing from the paper in this final form is a multiple time repeated experiment (3 to 5 times) and carefully looking at the zig-zag evolution in same measurement conditions (when repeated) and a discussion of the observed repeatability or changes based on the model, as the authors are referring to randomness of the switching process.

Response to comments:

Thanks for important suggestion. Actually, we have checked the reproducibility of experiments in the same measurement condition. The results are summarized in the following figure (Fig. S3(a)). The main characteristics, such as the biggest jumps near $\pm V_C$, followed by the zig-zag evolution are reproduced clearly among the 1st, 2nd and 3rd measurements. The voltage peak position showing the oscillating small jumps is not necessarily reproduced among three measurements but slightly scattered. It may be due to a possible randomness of the successive domain switching process as the reviewer suggested.

We have included the figure as Fig. S3(a) and commented this point in section I in the main text as well as in the supplementary materials III as follows.

Section I

This is further confirmed by the results in the supplementary material *III*, that the experimental observations are well reproduced by repeating the measurement for 3 times and the V_F positions corresponding to the biggest V_{int} jumps remain to be near $\pm V_C$ even by changing C_P from 15 nF down to 0.1 nF.

Supplementary materials III

Fig. S3 shows V_F - V characteristics **(a)** in FE/PE stack for three consecutive measurements in the same measurement condition and **(b)** in FE/PE stacks with four kinds of C_P . In all cases, the first and biggest V_F drop, corresponding to the first and biggest V_{int} jump, occurs near $\pm V_C$ of FE film. The small zig-zag following the biggest V_F drop is also reproduced well, although the zig-zag position is not necessarily overlapped among 1st, 2nd and 3rd measurements in (a). This may be due to a possible randomness in the consecutive domain switching.

Fig. S3 V_F - V characteristics (a) in FE/PE stack for repeating 3 times.

Reviewer #3 (Remarks to the Author):

The reviewer is satisfied with the responses to comments #1 and #3, but there remains some concerns about the response to comment #2.

Although the reviewer agrees that even with this leakage, the measured V_{int} and calculated V_{FE} could both be correct as the author indicates, it is still not clear how this leakage current would affect the observation of the apparent “negative capacitance” effect; i.e., in order to be consistent with the “negative capacitance” effect, one should observe that V_{int} rises faster than V_{APP} , but the author has failed to demonstrate whether the leakage current has erroneously caused this “negative capacitance” feature.

Response to comments:

Thanks for very suggestive comments and we are sorry for unclear explanation in the last revision. In this revision, let us explain the leakage current effect on the observed results more clearly by considering the time dependence of V_{int} in more detail. Since V_F drops correspond to V_{int} jumps, namely V_{int} rising faster than V_{APP} , let us focus on only V_F drop. The time (t) dependence of V_F is determined as

$$V_F(t) = V_{Fb} e^{-t/\tau},$$

$$\tau = R_F(C_P + C_F),$$

where V_{Fb} is V_F at the beginning of each measurement after setting a new V value, and τ is the time constant of V_F in the FE/PE system.

We have considered the reviewer#3’s comment #2 in the last review again. The reviewer might have a concern that a possible variation of the time constant in FE/PE system may result in the V_F drop in some cases. Yes, generally speaking it is expected that the resistance and/or capacitance may change with V_F in FE-CAP, so there may be a variation of τ . Thus, a possible V_F drop might be caused by the time constant variation in some cases, as schematically shown in Fig. R1.

Fig. R1 Schematics of possible effect of variation in time constant of FE/PE system on V_F .

However, the reviewer#3’s concern is irrelevant to the results observed in our experiments, as explained in the following.

If the V_F drop is caused by the variation of τ , it should become weaker or disappear when the time constant of the system becomes much larger. According to the above equation for the time constant, we

can increase the time constant τ by increasing C_p . However, Fig. S3(b) shows that both the biggest V_F drop and the small oscillating ones are not affected at all by increasing C_p from 0.1 nF to 15 nF, namely increasing τ by larger than two orders. The V_{int} gain factor corresponding to the biggest V_F drop is summarized as a function of C_p in Fig. R2(a). The gain factor with a larger C_p is even larger. This may be due to the larger P_1 switching with a larger C_p , and is consistent with our model. This result is contrary to that expected from the time constant variation, which may be originated from the nonlinear bias-dependence of leakage current and/or capacitance. Thus, it is concluded that both the biggest V_F drops and small zig-zag ones observed in our experiments can only be explained by the successive domain switching.

Fig. R2 Maximum V_{int} gain factor as a function of C_p in FE/PE system.

We have newly discussed this point in the text as follows.

In addition, a possible origin of the V_{int} variation in FE/PE system might be the time constant variation originated from the bias dependence of resistance and/or capacitance of FE-CAP. But, such a concern is irrelevant to the present experimental observations, because the results are not affected at all even in the FE/PE stacked systems with different time constants (with different C_p values).

In addition, the reviewer#3 might also have a concern that the polarization switching current may have an effect on V_{int} measurement. Polarization switching current is a transient displacement current and it does not affect the resistance of FE-CAP so much (Asit K. Maity et al, Jpn. J. Appl. Phys. 43, 7155 (2004)). In our DC measurement, we need not consider this effect seriously.

We hope the reviewer will be satisfied with our explanation on this issue. And, we would thank reviewer for the careful reviews and invaluable comments in the 1st, 2nd and 3rd runs.

Sincerely

Xiuyan Li

Shanghai Jiao Tong University

Reviewer #1 (Remarks to the Author):

The authors well modified the manuscript according to the comments. This reviewer suggests publication of the work in the present form.

REVIEWERS' COMMENTS:

Reviewer #1 (Remarks to the Author):

The authors well modified the manuscript according to the comments. This reviewer suggests publication of the work in the present form.

Response to comments:

We would thank reviewer #1 for the suggestion of publication of our paper and for the careful reviews and invaluable comments in the 1st, 2nd, 3rd and 4th runs of review.

Sincerely

Xiuyan Li

Shanghai Jiao Tong University